# Elucidating the design space of language models for image generation

Xuantong Liu[1]  Shaozhe Hao[2]  Xianbiao Qi[*3]  Tianyang Hu[†4]  Jun Wang[1]  Rong Xiao[3]  Yuan Yao[†1]

## Abstract

The success of large language models (LLMs) in text generation has inspired their application to image generation. However, existing methods either rely on specialized designs with inductive biases or adopt LLMs without fully exploring their potential in vision tasks. In this work, we systematically investigate the design space of LLMs for image generation and demonstrate that LLMs can achieve near state-of-the-art performance without domain-specific designs, simply by making proper choices in tokenization methods, modeling approaches, scan patterns, vocabulary design, and sampling strategies. We further analyze autoregressive models' learning and scaling behavior, revealing how larger models effectively capture more useful information than the smaller ones. Additionally, we explore the inherent differences between text and image modalities, highlighting the potential of LLMs across domains. The exploration provides valuable insights to inspire more effective designs when applying LLMs to other domains. With extensive experiments, our proposed model, **ELM** achieves an FID of 1.54 on 256×256 ImageNet and an FID of 3.29 on 512×512 ImageNet, demonstrating the powerful generative potential of LLMs in vision tasks.

## 1. Introduction

In the domain of artificial intelligence generated content (AIGC), text and image generation (Brown, 2020; Ho et al., 2022) represent the principal focal points. Despite their shared goal of content generation, these two modalities predominantly employ distinct methods. On the one hand, text generation is commonly facilitated by autoregressive (AR) language models, like LLaMA-3 (Touvron et al., 2023a) and GPT-4 (Achiam et al., 2023), which operate by predicting subsequent tokens based on preceding ones in a sequence. On the other hand, image generation predominantly utilizes diffusion models, such as Dall·E 3 (Betker et al., 2023) and Stable Diffusion v3 (Esser et al., 2024), which learn to gradually denoise images for all pixels simultaneously.

The recent success of large language models (LLMs) has bolstered the research community's confidence to achieve artificial general intelligence (AGI) and inspired their application in computer vision tasks, such as image (Esser et al., 2021; Yu et al., 2021; Tian et al., 2024) and video generation (Kondratyuk et al., 2024). A significant advantage of integrating LLMs into image generation is the ability to transfer established techniques from text-based applications, such as enabling undefined output length (Xiao et al., 2023) and speed-up strategies (Kwon et al., 2023). Moreover, the scalability of LLMs makes them the preferred foundation for building unified multi-modal models (Team et al., 2023; Kondratyuk et al., 2024). Nevertheless, LLMs for image generation remain underexplored. Recent efforts often rely on domain-specific strategies, such as sequencing visual tokens from low to high resolution (Tian et al., 2024), using continuous space modeling with diffusion loss (Li et al., 2024), or randomly permuting image tokens to utilize bidirectional information (Yu et al., 2024a). These approaches introduce inductive biases depending on domain knowledge, which may limit LLMs' true potential. Meanwhile, preliminary attempts (Esser et al., 2021; Chang et al., 2022; Yu et al., 2022; Sun et al., 2024; Tian et al., 2024; Yu et al., 2024b) take a simpler approach by discretizing images into token sequences with vector-quantization (VQ) autoencoders and training LLMs on token prediction objectives, leaving much of the design space unexplored.

In this study, we delve into the potential of language models for image generation tasks. Start with image tokenization, we compare VQGAN (Van Den Oord et al., 2017; Esser et al., 2021) and BAE (binary autoencoder) (Wang et al., 2023; Yu et al., 2023). Our comparison involving reconstruction ability, scalability, and generation performance shows that **BAE consistently outperforms VQGAN** across all dimensions. Despite this, current language model-based image generation methods largely rely on VQGAN (Yu

*Project lead, †Corresponding authors  [1]Hong Kong University of Science and Technology  [2]The University of Hong Kong  [3]Intellifusion  [4]National University of Singapore. Correspondence to: Xuantong Liu <xliude@connect.ust.hk>, Xianbiao Qi <qixianbiao@gmail.com>, Tianyang Hu <hutianyang.up@outlook.com>, Yuan Yao <yuany@ust.hk>.

*Proceedings of the 42nd International Conference on Machine Learning*, Vancouver, Canada. PMLR 267, 2025. Copyright 2025 by the author(s).

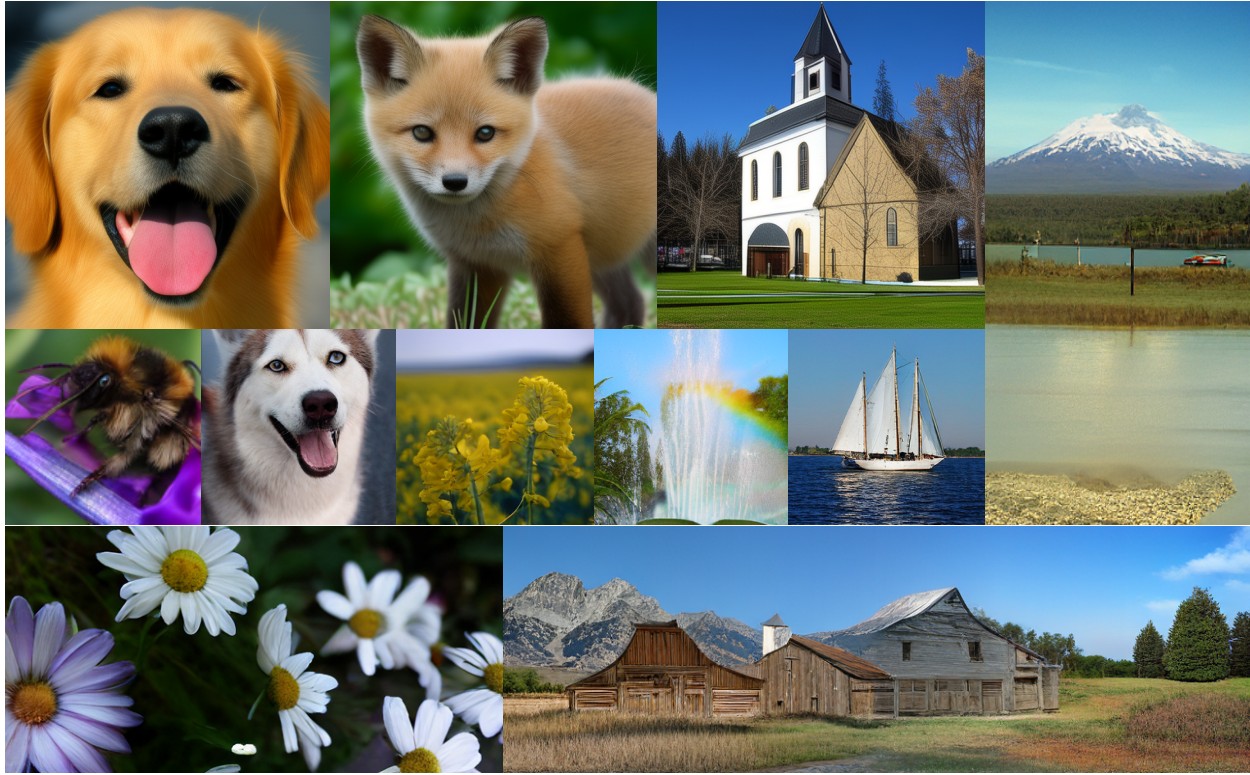

Figure 1: Generated samples by ELM-2B with 2-12 tokenizer trained on ImageNet. Top-row: generated $512 \times 512$ samples; mid-row: generated $256 \times 256$ samples; others: generated unfixed-size-samples with streaming behavior.

et al., 2022; Chang et al., 2022; Li et al., 2023; Sun et al., 2024). We believe that a more powerful tokenizer for images can lead to significantly better generation performance.

We then investigate the performance of two primary language modeling approaches for image generation: autoregressive (AR) models and masked language models (MLMs). Consistent with findings in the language domain (Henighan et al., 2020; Liao et al., 2020; Zhang et al., 2024; Chang & Bergen, 2024), **AR models** *demonstrate superior image generation ability and scalability compared to MLMs.* We explore different token scanning patterns for flattening 2D image tokens; *textbfrow-major raster scanning achieves the best performance.* By leveraging the flexibility of the binary codes produced by BAE, we investigate code decomposition strategies. Through our exploration, we refine the attempt in (Yu et al., 2023) and conclude that *splitting the original code into* **two subcodes** *reduces learning complexity, improves performance, and saves computational costs.* During inference, we explore the optimal sampling strategies for AR models and MLMs, highlighting the importance of *randomness* in generating realistic images.

We analyze how AR models learn to generate images by examining attention scores across different layers and model sizes. Our findings indicate that AR models effectively learn

the importance of local information for image generation. However, larger models also capture *global information*, which is more difficult for smaller models to learn, helping to explain the performance improvements observed with increasing model size. Additionally, we observe that image tokens exhibit *greater randomness and lack the inherent orderliness* compared to text tokens, showing the fundamental difference between the language and vision domains, which presents challenges for training image generation with the token prediction objective. LLMs achieving high image generation performance highlights their potential across domains.

Our research deepens the understanding of the LLM's capability and behavior in vision generation. The insights can contribute to the design of more efficient and unified large models. In conclusion, our main contributions include:

- We thoroughly examine two prevalent language modeling methods, AR models and MLMs, within the realm of image generation. Our findings suggest that AR holds greater potential in image generation.

- Leveraging BAE as the image tokenizer, our results reveal that appropriate vocabulary decomposition helps improve performance and reduce computational costs.

- We show that AR models can learn effective image patterns without inductive bias, identify distinct patterns across model sizes, and offer a concise explanation of the scaling law.

- Combining all key ingredients of the design space, we reach a strong **E**lucidated **L**anguage model for i**M**age generation, termed as **ELM**, and achieve top-tiered performance on the ImageNet benchmark.

## 2. Preliminary

### 2.1. Image Tokenization

Image tokenization typically involves an encoder ENC, a quantizer QUANT, and a decoder DEC. Given an image $\boldsymbol{x} \in \mathbb{R}^{H \times W \times 3}$, ENC encodes it to latent variables $\boldsymbol{z} = \text{ENC}(\boldsymbol{x}) \in \mathbb{R}^{\frac{H}{f} \times \frac{W}{f} \times D}$. Each spatial vector $\boldsymbol{z}_{ij}$ in $\boldsymbol{z}$ is then quantized to discrete code $\boldsymbol{q}_k$. Let the quantized latent be denoted as $\boldsymbol{z}_q$, $\hat{\boldsymbol{x}} = \text{DEC}(\boldsymbol{z}_q)$ denotes the reconstructed image (Van Den Oord et al., 2017; Razavi et al., 2019; Esser et al., 2021; Yu et al., 2023; Hu et al., 2023). All the codes form a codebook $\mathcal{Q} = \{\boldsymbol{q}_k\}_{k=1}^K \subset \mathbb{R}^D$, which can be viewed as the "vocabulary" if we regard the image as a special kind of language. A sequence of tokens $\boldsymbol{q} = (\boldsymbol{q}_1, \boldsymbol{q}_2, ..., \boldsymbol{q}_L)$, where $L = \frac{H}{f} \times \frac{W}{f}$, is obtained by reshaping $\boldsymbol{z}_q$.

**VQGAN** trains the codebook $\mathcal{Q}$ alongside ENC and DEC. In this method, each spatial latent vector $\boldsymbol{z}_{ij} \in \mathbb{R}^D$ "looks up" the nearest code $\boldsymbol{q}_k$ by minimizing the Euclidean distance (Razavi et al., 2019; Esser et al., 2021):

$$\boldsymbol{z}_q = \left( \arg \min_{\boldsymbol{q}_k \in \mathcal{Q}} \|\boldsymbol{z}_{ij} - \boldsymbol{q}_k\| \right) \in \mathbb{R}^{\frac{H}{f} \times \frac{W}{f} \times D}. \quad (1)$$

**BAE** discretizes the scalar value at each position of $\boldsymbol{z}_{ij}$, converting it to a binary value (Fajtl et al., 2020; Wang et al., 2023; Yu et al., 2023). Specifically, suppose the latent vector $\boldsymbol{z}_{ij} \in \mathbb{R}^D$ is normalized, then each value $z^d, d \in \{1, ..., D\}$ at the $d$-th position of $\boldsymbol{z}_{ij}$ is obtained by:

$$z_q^d = \text{sign}(z^d) = \begin{cases} 0, & \text{if } z^d < 0.5, \\ 1, & \text{otherwise.} \end{cases} \quad (2)$$

In this way, the codebook is structured within a binary latent space, with $K = 2^D$. This method is also named "look-up free" quantization (LFQ) (Yu et al., 2023) because the code can be directly converted into an index by transforming the binary value into a decimal value. The sign function can be replaced by Bernoulli sampling, then $\boldsymbol{z}_q = \text{Bernoulli}(\boldsymbol{z})$ (Wang et al., 2023).

### 2.2. Modeling Methods

**Autoregressive (AR) Models** involve a sequence of discrete tokens $\boldsymbol{q} = (\boldsymbol{q}_1, \boldsymbol{q}_2, ..., \boldsymbol{q}_L)$, where each token $\boldsymbol{q}_l$ is

drawn from a vocabulary $\mathcal{Q}$ of size $K$. The AR model assumes that the probability of the current token $\boldsymbol{q}_l$ depends only on its preceding tokens $(\boldsymbol{q}_1, \boldsymbol{q}_2, ..., \boldsymbol{q}_{l-1})$, framing the generation task as a 'next-token' prediction, with a unidirectional attention mechanism. Specifically, the network learns the probability $p(\boldsymbol{q}) = \Pi_{l=1}^L p(\boldsymbol{q}_l \mid \boldsymbol{q}_1, \cdots, \boldsymbol{q}_{l-1})$, with the loss function:

$$\mathcal{L}_{\text{ar}} = -\mathbb{E}_{\boldsymbol{x} \sim p(\boldsymbol{x})} [\log p(\boldsymbol{q})] \quad (3)$$

**Masked Language Models (MLMs)** involves a binary mask $\boldsymbol{m} = [m_l]_l^L$ to replace a subset of tokens in $\boldsymbol{q}$ with [MASK], then predict the masked tokens based on the unmasked ones from both directions. Specifically, if $m_i = 1$, $\boldsymbol{q}_i$ is replaced by [MASK]; otherwise, it remains unchanged. Denote $\boldsymbol{q}_M$ the result after applying mask $\boldsymbol{m}$ to $\boldsymbol{q}$. Hence, these models optimize the following loss function:

$$\mathcal{L}_{\text{mlm}} = -\mathbb{E}_{\boldsymbol{x} \sim p(\boldsymbol{x})} \left[ \sum_{\forall l \in [0,L], \; m_l = 1} \log p(\boldsymbol{q}_l \mid \boldsymbol{q}_M) \right] \quad (4)$$

## 3. Elucidating the Design Space of language models for image generation

In this section, we comprehensively explore the design space of adopting language models for vision generation, including the tokenizer choice, modeling choice, scan pattern choice, model scalability analysis, vocabulary decomposition strategy, and sampling strategy, and conduct comprehensive experiments on the 256×256 ImageNet (Deng et al., 2009) benchmark, mainly use Fréchet Inception Distance (FID) (Heusel et al., 2017) as the evaluation metric.

### 3.1. Tokenizer Choice: VQGAN v.s. BAE

Table 1: Comparison of VQGAN and BAE[1]. Tokenizers are trained on the ImageNet. Generation models are L sized.

|         | **VQGAN** | | **BAE** | | | |
|---------|-----------|-------|----------|----------|----------|----------|
| $K$     | 1024      | 16384 | $2^{16}$ | $2^{20}$ | $2^{24}$ | $2^{32}$ |
| rFID    | 10.54     | 7.41  | 3.32     | 2.24     | 1.77     | 1.68     |
| gFID(M.)| 11.21     | 7.81  | 3.96     | 3.65     | 3.91     | -        |
| gFID(A.)| 9.68      | 6.71  | 2.78     | 2.46     | 2.68     | -        |

In VQGAN, "code collapse" is a critical issue where a large portion of the codebook remains unused as the codebook size increases, severely limiting the model's efficiency and scalability (Zhu et al., 2024; Baykal et al., 2024). This problem does not occur in BAE, where discrete codes are generated using scalar quantization (Mentzer et al., 2023).

---

[1]In this work, we adopt a down-sampling factor of $f = 16$ for image tokenizers.

This approach guarantees 100% code utilization (**Appendix A.5.2**) and achieves better reconstruction capabilities (rFID). We also compare the generation capabilities (gFID) of both tokenizers using different codebook sizes ($K$) and generation modeling approaches, *i.e.*, MLM and AR. Among all the results, BAE consistently demonstrates superior performance (**Table** 1). *Based on the above reasons, we build our generation model on BAE tokenizer instead of VQGAN.*

For BAE, we observe that the introduction of *Bernoulli Sampling* during quantization improves generation performance (**Table** 10). Incorporating the probabilistic element reduces the model's sensitivity to prediction errors (Englesson & Azizpour, 2021), leading to a more robust generation.

### 3.2. Modeling Method Choice: AR *v.s.* MLM

In this subsection, we evaluate the performance of AR and MLM in image generation with the same BAE tokenizer with a vocabulary size of $2^{16}$ and training strategy.

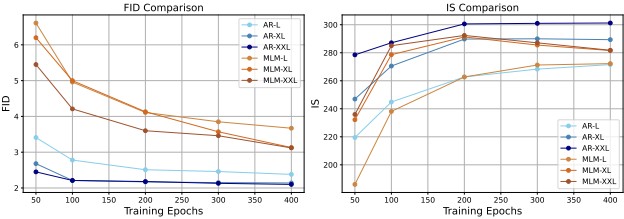

Figure 2: Comparison of AR and MLM on image generation with 50,000 generated samples. AR consistently outperforms MLM across various model sizes.

**Figure** 2 presents the FID score and Inception Score (IS) (Salimans et al., 2016) over the training epochs for both AR and MLM. The results show that *AR consistently outperforms MLM across various model sizes.* Additionally, *AR exhibits higher training efficiency* compared to MLM, particularly as the model size increases. Research in the language domain has widely recognized that AR models possess greater generative capabilities than MLMs, particularly as model scales increase (Radford et al., 2019; Raffel et al., 2020; Henighan et al., 2020). Our findings align with these works. Besides, for MLM-XL and MLM-XXL, a clear divergence between FID and IS is observed in the later stages of training, where FID continues to improve, while IS declines. Studies point out that when models overfit to generate highly realistic samples (low FID), they may sacrifice diversity, which negatively impacts IS (Chong & Forsyth, 2020; Benny et al., 2021). This issue does not occur with AR models, further highlighting the superiority of AR over MLM in maintaining both quality and diversity.

### 3.3. Scan Pattern Choice

For AR modeling, converting 2D image tokens into 1D sequences involves the challenge of choosing an appropriate

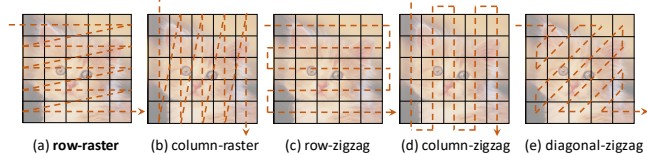

Figure 3: Scan pattern choices of converting 2D image tokens into 1D sequences.

scan pattern. In this study, we explore five possible options: row-raster, column-raster, row-zigzag, column-zigzag, and diagonal-zigzag, as illustrated in **Figure** 3. The results (**Table** 2) reveal the various scanning patterns achieve good performance, demonstrating that AR models can effectively learn image generation regardless of the scanning pattern. In our subsequent exploration, *for AR models, we adopt the **row-major raster** scanning method*, which is the most natural one and yields the best results among all scanning methods.

Table 2: Comparison of different scan pattern choices. Tokenizer: BAE; AR model size: L.

| pattern | (a) | (b) | (c) | (d) | (e) |
|---------|-----|-----|-----|-----|-----|
| gFID | **2.47** | 2.88 | 2.62 | 2.79 | 2.71 |

### 3.4. Learning and Scaling Behavior

To further understand the model's learned patterns, we visualize the attention maps of AR models with different sizes.

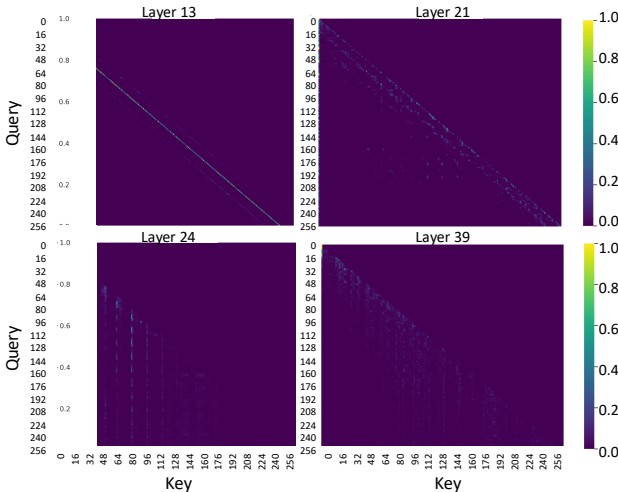

Figure 4: Visualization of average attention score of head 0 in AR models over 100 images. We show the results of L (top row), and XL (bottom row) models with the BAE 2-10 tokenizer, respectively. Both models effectively learned to focus on *localized information*. However, the XL model learns to capture richer *global information*.

The results reveal that the attention mechanism is primarily focused on *local regions* of the image, indicating that the AR transformer models effectively learn the importance of local patterns for image generation (Vaswani et al., 2017). This finding is notable because the model was trained without any inductive biases tailored to image data, highlighting the strong capability of AR transformer models across different domains. Besides, for the attention pattern, models of varying sizes showed subtle differences: the L-sized model mainly focused on local information, struggling to capture long-term relations. In contrast, larger models (XL) with more layers exhibited longer-range attention in certain layers, suggesting they also learn global features (**Figure** 4) This capability contributes to improved generation performance, as evidenced by better visual quality and lower FID scores.

Clearly, the *scaling law* (Henighan et al., 2020; Kaplan et al., 2020) holds for AR models in image generation tasks, as reflected in *lower training loss* (**Figure** 15), *improved generation performance*, and an enhanced ability to capture *global information* as the model size increases.

### 3.5. Vocabulary Design

The vocabulary size $K$ in the BAE tokenizer is determined by the code dimension $D$, *ie*, $K = 2^D$. When the vocabulary size exceeds a certain threshold, such as $2^{16}$ (*ie*, 65,536), next-token prediction becomes significantly more challenging (Ali et al., 2023) and may even become infeasible due to memory constraints. Despite these limitations, the tokenizer's effectiveness largely depends on the code dimension, increased code dimension results in an improved reconstruction ability (**Figure** 5). Recent research also indicates that a stronger tokenizer leads to a better generation performance in AR models (Tao et al., 2024). Therefore, addressing the challenge posed by larger codebook sizes (greater $D$) is crucial.

Fortunately, the flexibility of binary-quantized codes allows us to decompose each code into multiple subcodes (Yu et al., 2023), resulting in feasible vocabularies. For instance, an 8-bit code like $[1, 0, 1, 0, 0, 0, 1, 1]$ can be split into two 4-bit codes: $[1, 0, 1, 0]$ and $[0, 0, 1, 1]$. As a result, we convert the embedding matrix from a size of $2^8 \times D_{feature}$ into two matrices of size $2^4 \times D_{feature}$, where $D_{feature}$ is the feature dimension within the AR model. The final embedding is achieved by concatenating the two indexed embeddings and applying a projection to map the dimension back to $D_{feature}$. Separate prediction heads are applied.

We conduct experiments using AR models with BAE that have varying code dimensions ($D = 16, 20, 24$, and $32$). Quantizers with and without code decomposition are viewed as distinct tokenizers; for example, for $D = 16$, "1-16"

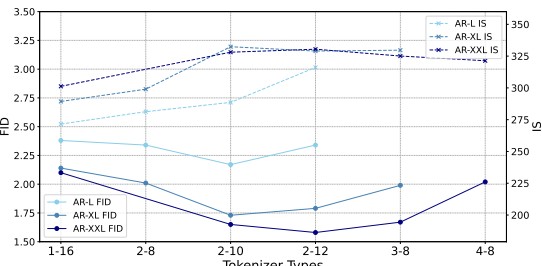

Figure 5: AR model performance with different BAE.

means the original tokenizer, and "2-8" denotes the code is split into two 8-bit subcodes. Several key insights are revealed through our design:

• **Optimal decomposition.** A decomposition into *two subcodes* is generally optimal, which also *reduces computational costs*, leading to more *efficient and effective generation* (detailed result in **Table** 12). Dealing with two subvocabularies of a smaller size leads to a more manageable set of possible outcomes, largely reducing the cognitive load on the model (Ali et al., 2023; Yang, 2024). Meanwhile, further increasing the number of subcodes significantly raises the prediction complexity, and the model struggles to optimize across three or more classification tasks simultaneously (Limisiewicz et al., 2023). (An increasing training loss was observed when moving from tokenizers 2-8 to 3-8 and 4-8 in XL and XXL models, as in **Figure** 14).

• **Vocabulary complexity and model capacity.** Larger code dimensions also introduce more complex vocabularies, making it harder for the model to predict the next token. Clearly, *more complex tokenizers require more powerful models* for effective learning (Tao et al., 2024). For example, the 2-10 tokenizer is optimal for L and XL models, while the 2-12 tokenizer performs best with the XXL model.

These findings demonstrate the trade-offs between model scale, vocabulary complexity, and decomposition strategies, highlighting the potential of the AR model's ability to effectively handle complex tokenization while maintaining high performance across model scales.

### 3.6. Sampling Strategy

Sampling strategy plays a crucial role in vision generation for both diffusion models (Karras et al., 2022; Ma et al., 2023) and language models (Chang et al., 2022; Sun et al., 2024). In this subsection, we thoroughly explore the sampling strategies for both AR and MLM, including classifier-free guidance (CFG) (Ho & Salimans, 2022) scale, the introduction of randomness, and the number of generation iterations for the MLMs.

First, regarding the CFG scale, a *linearly increased CFG scale* shows the best performance among various scheduling

methods (**Table** 13). Secondly, a high degree of randomness is crucial during the sampling process (**Figure** 6) for both methods. Regarding the introduction of randomness, for the AR model, randomness primarily derives from the top-$k$ filter used when selecting next-token indices based on their confidence scores; a larger $k$ introduces more randomness. For the MLM, randomness stems from the coefficient $\tau$ of Gumbel noise added to the [MASK] token predicted confidence; a larger $\tau$ results in greater randomness. Visual results (**Figure** 17, 18 and 19) further reflect the influence of CFG and top-$k$ sampling for AR models and the difference from language domain.

Moreover, empirical results in (**Figure** 6) also indicate when *model size and vocabulary increase, the need for randomness diminishes*, indicating that larger models are capable of capturing a *broader range of patterns* and making *more accurate predictions*. This observation aligns with the attention and scalability analysis discussed earlier, where larger models demonstrated enhanced capacity to manage both local and global information, reducing the need for stochasticity to generate realistic samples.

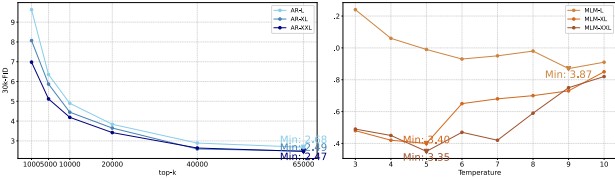

Figure 6: The influence of 'randomness' during sampling. Left: top-$k$ on AR; right: $\tau$ on MLM.

Finally, in the context of MLM-based image generation, the number of sampling iterations typically ranges from 1 to the total sequence length. Our experiments indicate that approximately 10 iterations strike an effective balance between generation quality and computational efficiency (**Figure** 20), highlighting the sampling efficiency advantage of MLMs over their AR counterparts.

### 3.7. ELM Model

After thoroughly exploring the design space of language models for image generation, via combining the better design trick, we reach our final **E**lucidated **L**anguage model for i**M**age generation (**ELM**). ELM adopts *BAE* as the image tokenizer and *AR* as the modeling method with a *row-raster* scanning pattern; and splits the quantized image code into *two subcodes*. The vocabulary should align with the model's capacity. Larger vocabularies require more powerful models to handle the next-token prediction task (2-12 tokenizer for ELM-XXL), while smaller models perform better with simpler vocabularies (2-10 tokenizer for ELM-L and XL). For sampling strategy, we choose a *high randomness* and a *linear* CFG scale. In total, we construct four ELM versions:

ELM-L (2-10), ELM-XL (2-10), ELM-XXL (2-12), and ELM-2B (2-12), with parameters ranging from 315M to 1.9B.

## 4. Experimental Results

### 4.1. Conditional Image Generation

In this section, we compare our ELM models with other popular image generation models on the 256×256 ImageNet. We compare with Diffusion Models including DiT (Peebles & Xie, 2023) and SiT (Ma et al., 2024); Masked Language Models such as MaskGIT (Chang et al., 2022); Autoregressive models, including VQGAN (Esser et al., 2021) and LlamaGen (Sun et al., 2024); Visual Autoregressive Model (VAR) (Tian et al., 2024) and Masked Autoregressive Model (MAR) (Li et al., 2024). The comparison result is presented in **Table** 3.

Table 3: Comparison results of class-conditional image generation on 256×256 ImageNet. * indicates that the model generates samples at a resolution of 384×384, which are then resized to 256×256. -re denotes rejection sampling is used.

| Type | Model | Para. | FID↓ | IS↑ | Pre.↑ | Rec.↑ |
|---|---|---|---|---|---|---|
| **Diff.** | DiT-L/2 | 458M | 5.02 | 167.2 | - | - |
| | DiT-XL/2 | 675M | 2.27 | 278.2 | 0.83 | 0.57 |
| | SiT-XL/2 (ODE) | 675M | 2.15 | 258.1 | 0.81 | 0.60 |
| | SiT-XL/2 (SDE) | 675M | 2.06 | 277.5 | 0.83 | 0.59 |
| **MLM** | MaskGIT | 227M | 6.18 | 182.1 | 0.8 | 0.51 |
| | MaskGIT-re | 227M | 4.02 | 355.6 | - | - |
| **AR** | VQGAN | 227M | 18.65 | 80.4 | 0.78 | 0.26 |
| | VQGAN-re | 1.4B | 5.20 | 280.3 | - | - |
| | LlamaGen-L | 343M | 3.81 | 248.3 | 0.83 | 0.52 |
| | LlamaGen-XL | 775M | 3.39 | 227.08 | 0.81 | 0.54 |
| | LlamaGen-XXL | 1.4B | 3.09 | 253.61 | 0.83 | 0.53 |
| | LlamaGen-3B | 3.1B | 3.05 | 222.33 | 0.80 | 0.58 |
| | LlamaGen-3B* | 3.1B | 2.18 | 263.33 | 0.81 | 0.58 |
| **VAR** | VAR-d16 | 310M | 3.30 | 274.4 | 0.84 | 0.51 |
| | VAR-d20 | 600M | 2.57 | 302.6 | 0.83 | 0.56 |
| | VAR-d24 | 1.0B | 2.09 | 312.9 | 0.82 | 0.59 |
| | VAR-d30 | 2.0B | 1.97 | 334.7 | 0.81 | 0.61 |
| | VAR-d30-re | 2.0B | 1.80 | **356.40** | 0.83 | 0.57 |
| **MAR** | MAR-B | 208M | 2.31 | 281.7 | 0.82 | 0.57 |
| | MAR-L | 479M | 1.78 | 296.0 | 0.81 | 0.60 |
| | MAR-H | 943M | 1.55 | 303.7 | 0.81 | 0.62 |
| **AR** | **ELM**-L (2-10) | 315M | 2.17 | 288.59 | 0.82 | 0.55 |
| | **ELM**-XL (2-10) | 757M | 1.79 | 332.99 | 0.80 | 0.59 |
| | **ELM**-XXL (2-12) | 1.4B | 1.58 | 330.43 | 0.80 | 0.60 |
| | **ELM**-2B (2-12) | 1.9B | **1.54** | 332.69 | 0.81 | 0.60 |

We generate 50,000 samples to evaluate performance using FID, IS, Precision, and Recall following (Dhariwal & Nichol, 2021). Implementation details can be found in the **Appendix** A.5. Our method (ELM) exhibits scaling law behavior, with performance improving as model size increases.

Our tokenizers, *ie,* BAE, are only trained on ImageNet, we believe further training on larger datasets, like OpenImages (Kuznetsova et al., 2020), would enhance the tokenizer and further boost the generation capability of our ELMs.

### 4.2. Efficient transfer to higher resolution

We further train our model on 512×512 ImageNet to validate the effectiveness of the AR mechanism in seamlessly transferring from low-resolution (short-sequence) to high-resolution (long-sequence) image generation tasks. Specifically, we fine-tune a model pre-trained on 256×256 ImageNet directly on 512×512 ImageNet. Remarkably, we found that the model achieved excellent performance with only a few epochs of fine-tuning, effectively transitioning from generating 256 tokens to generating 1024 tokens. This significantly improves the efficiency of training image generation models on high-resolution datasets, as the model can first be trained on smaller resolutions and then fine-tuned with minimal effort for larger resolutions. The results on 512×512 ImageNet are presented in **Table 4**, with 50 epochs of fine-tuning.

Table 4: Comparison results of class-conditional image generation on 512×512 ImageNet.

| Model | Para. | Steps | FID↓ | IS↑ | Pre.↑ | Rec.↑ |
|---|---|---|---|---|---|---|
| DiT-XL/2 | 675M | 3000k | 3.04 | 240.82 | 0.84 | 0.54 |
| MaskGIT | 227M | 1500k | 7.32 | 156.0 | 0.78 | 0.50 |
| **ELM**-L (2-8) | 312M | 250k | 4.82 | 246.87 | 0.81 | 0.59 |
| **ELM**-XL (2-12) | 757M | 250k | 3.29 | 321.21 | 0.81 | 0.60 |

### 4.3. Visualization of the scaling law

According to the scaling law of ELM transformers, both loss and performance improve as training data and model parameter size increase. In our experiments, although the original image data (ImageNet) remains unchanged, the token set effectively scales up with different BAEs. We present generated samples using different sizes of ELM models (L, XL, XXL) and tokenizers (1-16, 2-10, 2-12) to illustrate the scaling behavior of ELM models in image generation. Following (Tian et al., 2024), we maintain the same seed and teacher-forced initial tokens across models. The results in **Figure** 7 clearly demonstrate performance improvements as both the token set and model size scale up.

### 4.4. Generalization of ELM

To further demonstrate the generalization and applicability of ELM, we evaluate its ability to generate samples from novel classes and perform image editing tasks, with the results in **Appendix** A.2. Additionally, we test our method on specific datasets, including a human face dataset and a texture dataset, to highlight its versatility; the results are presented in **Appendix** A.3. These experiments effectively showcase the robustness and adaptability of our approach.

## 5. Related Work

**Large Language Models** Language models are foundational tools in natural language processing, designed to predict the likelihood of sequences of words or tokens, using Transformer architectures with self-attention mechanism (Vaswani et al., 2017). There are two primary types: autoregressive (AR) models, like GPT (Radford et al., 2019; Brown, 2020; Achiam et al., 2023), LLaMA (Touvron et al., 2023a;b; Dubey et al., 2024), etc., which generate text one token at a time in a left-to-right fashion, and masked language models (MLM), such as BERT (Devlin, 2018), T5 (Raffel et al., 2020), etc., which predict masked tokens within a sequence using bidirectional context. AR models are particularly effective for text generation due to their sequential nature, while MLMs are better suited for representation learning by leveraging global context (Chang & Bergen, 2024). The scaling law (Henighan et al., 2020; Kaplan et al., 2020), which describes the relationship between the growth of model parameters, dataset sizes, computational resources, and performance improvements, highlights the immense potential of AR models.

**Vision Generation** Vision generation is a key focus in the current AIGC field, primarily relying on diffusion probabilistic models, which generate images by progressively denoising a random Gaussian noise (Song et al., 2020; Peebles & Xie, 2023; Chen et al., 2023). Transformer architectures are also the dominant backbone in these tasks (Yu et al., 2021; Peebles & Xie, 2023). Language models have also been applied to vision tasks. For instance, (Chang et al., 2022), (Li et al., 2023), and (Chang et al., 2023) use bidirectional MLMs for image generation, while (Esser et al., 2021), (Yu et al., 2022), and (Sun et al., 2024) employ AR models. Specifically, (Tian et al., 2024), (Li et al., 2024), and (Yu et al., 2024a) incorporate domain-specific designs to adapt AR models for image generation tasks. Moreover, AR models offer a path toward developing unified models for general artificial intelligence across different modalities, as seen in systems like Gemini (Team et al., 2023) and Chameleon (Team, 2024). Apart from existing works, our study emphasizes exploring the inherent potential of the vanilla AR paradigm in the vision domain with minimal modifications. We also investigate the learning behavior of language models, underscoring the versatility and potential of AR models across different modalities.

## 6. Discussion and Conclusion

**Token Randomness in Image Data** While images can be discretized and treated as token sequences, the inherent differences between vision and text still exist. In our exper-

Scaling up the tokenizer

Scaling up the model size

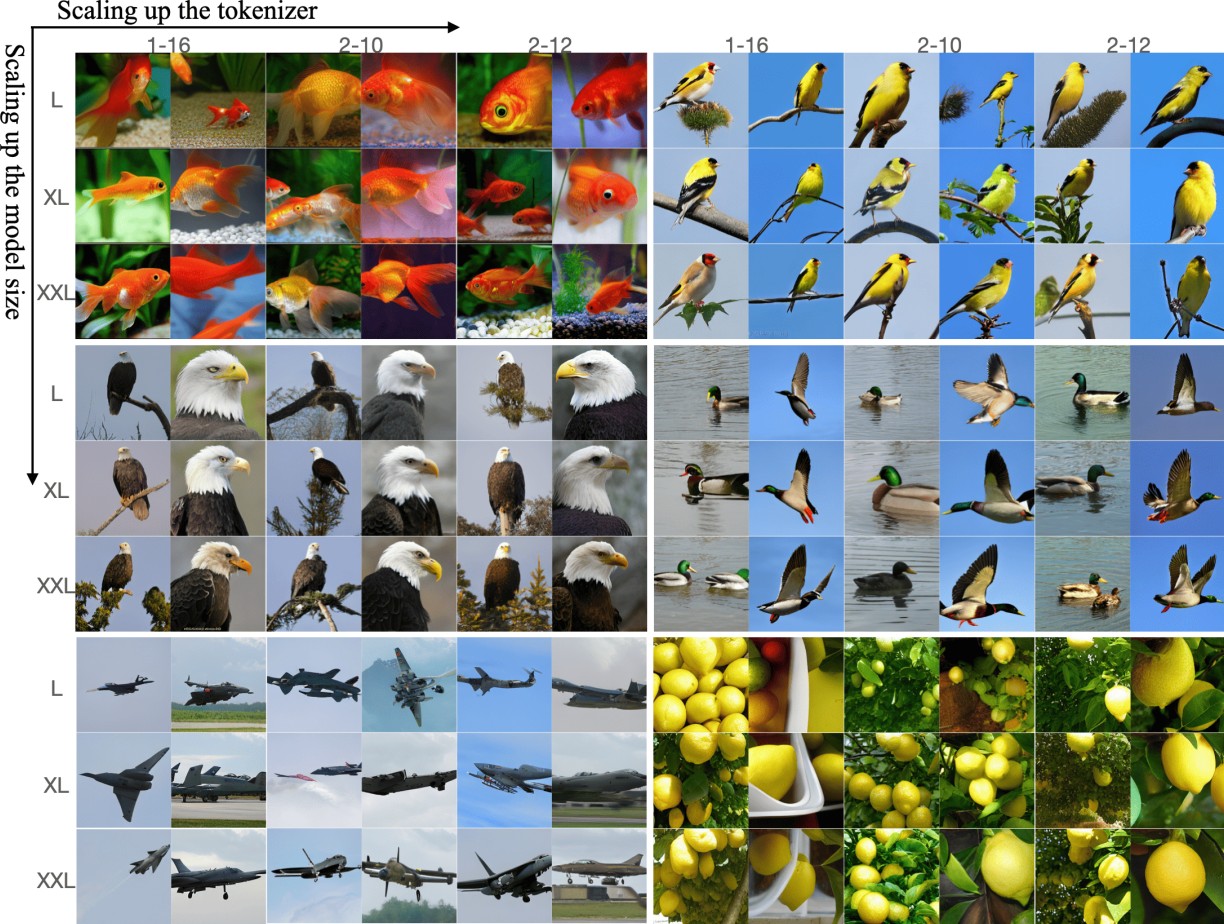

Figure 7: Scaling up behavior of tokenizer and model size. From left to right and top to bottom, there is a trend of improved image detail and structure. It reflects the enhanced generation ability that comes with a more refined tokenizer and a larger model.

iments, we observe that the training loss did not converge well with the token-prediction paradigm, whether AR or MLM, on image tokens, a similar result is also presented in Henighan et al. (2020). However, the models can still generate high-quality images with a low FID, indicating that they have learned sufficient patterns for image generation, although the training loss remains high.

Our analysis shows that image tokens exhibit a distribution much closer to a *random, uniform distribution* when compared to language tokens, and they exhibit *a lack of orderliness* based on bigram distribution and n-gram models' perplexity (**Appendix** A.4). These observations lead to several key implications. First, it suggests that image data *lacks the inherent structure and sequential order* that typically present in language data, implying that image generation is less dependent on strict sequential patterns and more on *local patterns* relevant to visual reconstruction (Ulyanov et al., 2018), aligning with our earlier analysis of the learning behavior of AR models in image generation.

Second, a token distribution close to uniform indicates that the generation task has a *higher tolerance for errors* (Zhang et al., 2016; Arpit et al., 2017). Since all tokens are nearly equally probable, the model can afford to make less precise token predictions without significantly impacting the quality of the generated output. This characteristic explains why our model, despite its high training loss, can still generate high-quality images and underscores the importance of incorporating randomness during inference.

**Conclusion** In this work, we thoroughly investigate the use of language models for image generation. We elucidate the design space of language models for vision generation, including tokenizer choice, model choice, scan pattern choice, model scalability, vocabulary design, and sampling strategy through extensive comparative experiments. Through our analysis, we have the following findings: (1) *BAE* demonstrates superior performance as an image tokenizer compared to traditional VQGAN approaches; (2) *AR models* consistently outperform MLMs and show a strong

scaling law, (3) *row-major raster* scanning performs best for flattening image tokens, (4) *larger vocabulary size* and a *decomposition* design benefit the image generation, (5) sampling strategies should also allow for *greater randomness* and a *linear CFG scale*. By combining these designs, we reach our final ELM model with near SOTA performance on ImageNet. We hope this work will motivate further usage of the AR model across other domains.

## Impact Statement

This research demonstrates the potential of language models in image generation, advancing AIGC and multimodal AI. The work has broad applications in creativity, education, and visualization, but raises ethical concerns, such as misuse in generating harmful content or biases in outputs. Future efforts should focus on responsible development and safeguards to ensure positive societal impact while fostering innovation in generative AI.

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

# A. Appendix

## A.1. Selected samples

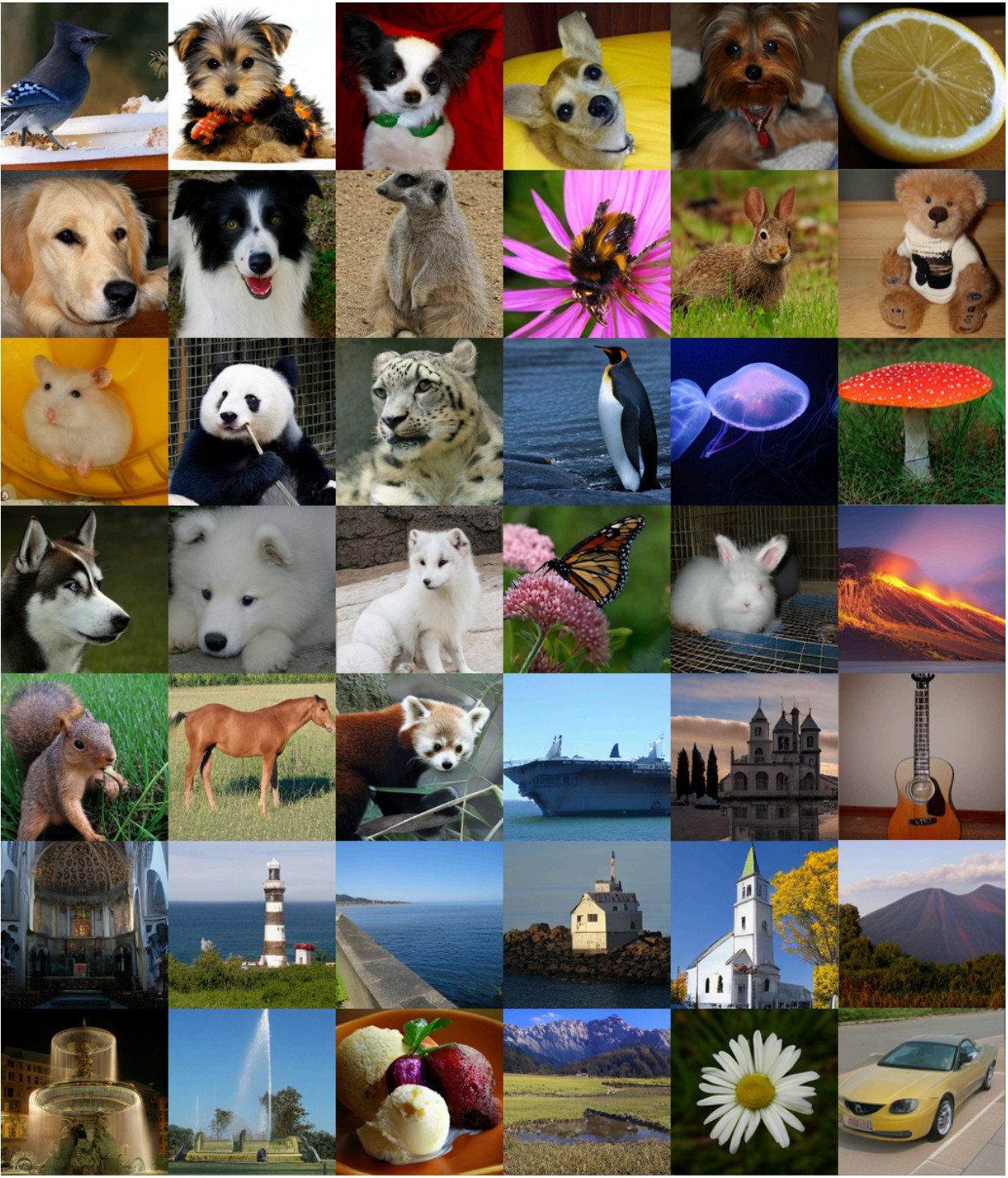

Figure 8: Selected 256×256 samples in different classes with ELM-2B (2-12).

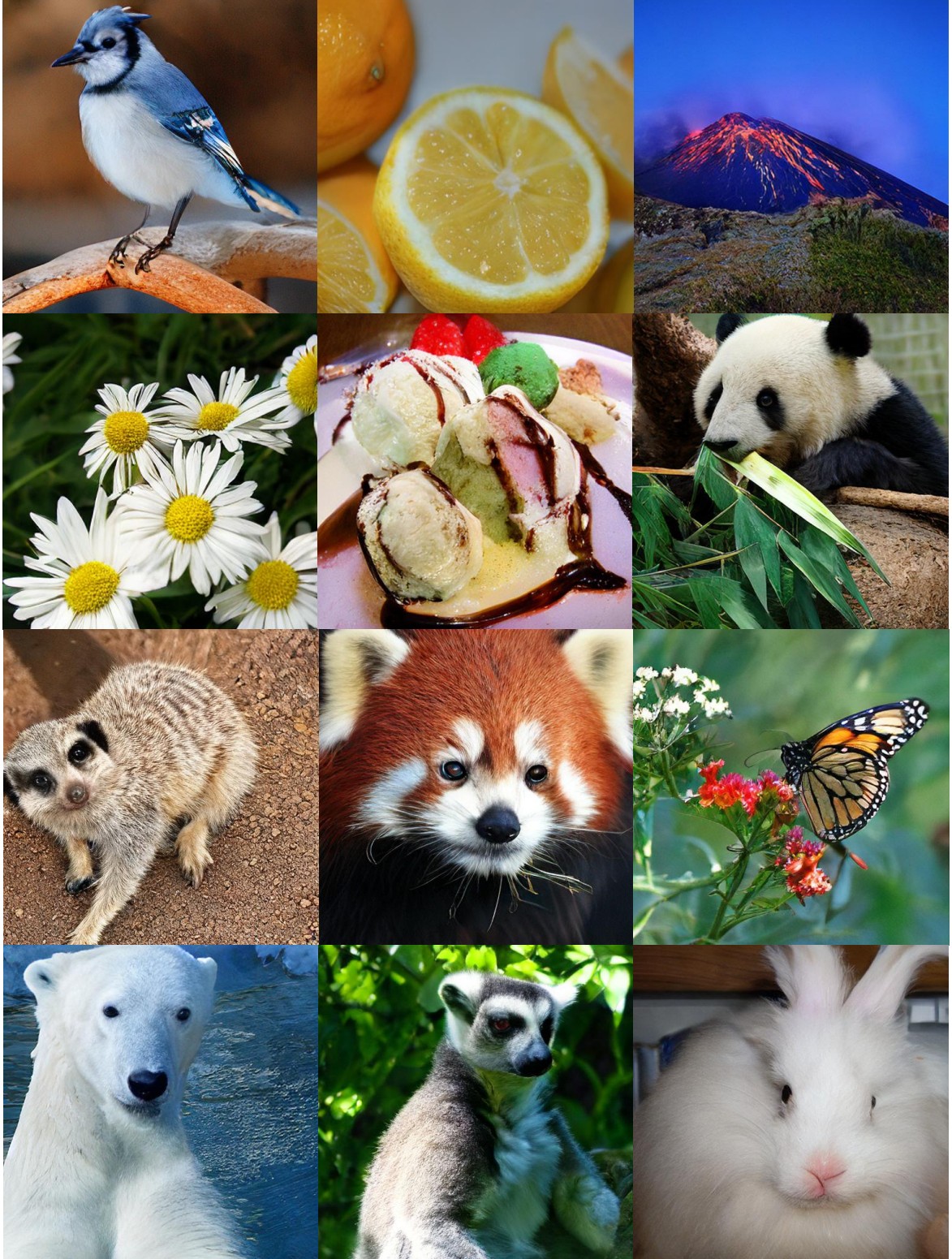

Figure 9: Generated $512 \times 512$ images.

## A.2. Generalization Ability of ELM models

We evaluated the model's performance on generating images with interpolated class conditions, specifically, $\alpha A + (1-\alpha)B$, where $A$ and $B$ are two distinct class labels, $\alpha \in [0, 1]$. This approach effectively tests how the model learns and adapts to conditions, especially under complex scenarios. The results show that the model effectively learns the conditional information, rather than simply memorizing it. Interestingly, when the interpolated classes share similarities, such as a golden retriever and a husky, the model generates images that blend features of both classes when $\alpha$ is around 0.5. In contrast, for unrelated classes like a sorrel and a beer bottle, the generated images only reflect the features of the class with the higher weight.

We also evaluate the image editing task by selecting specific regions of the original image and transforming them into other objects based on class conditions. The results further emphasize the flexibility of ELM across diverse application scenarios.

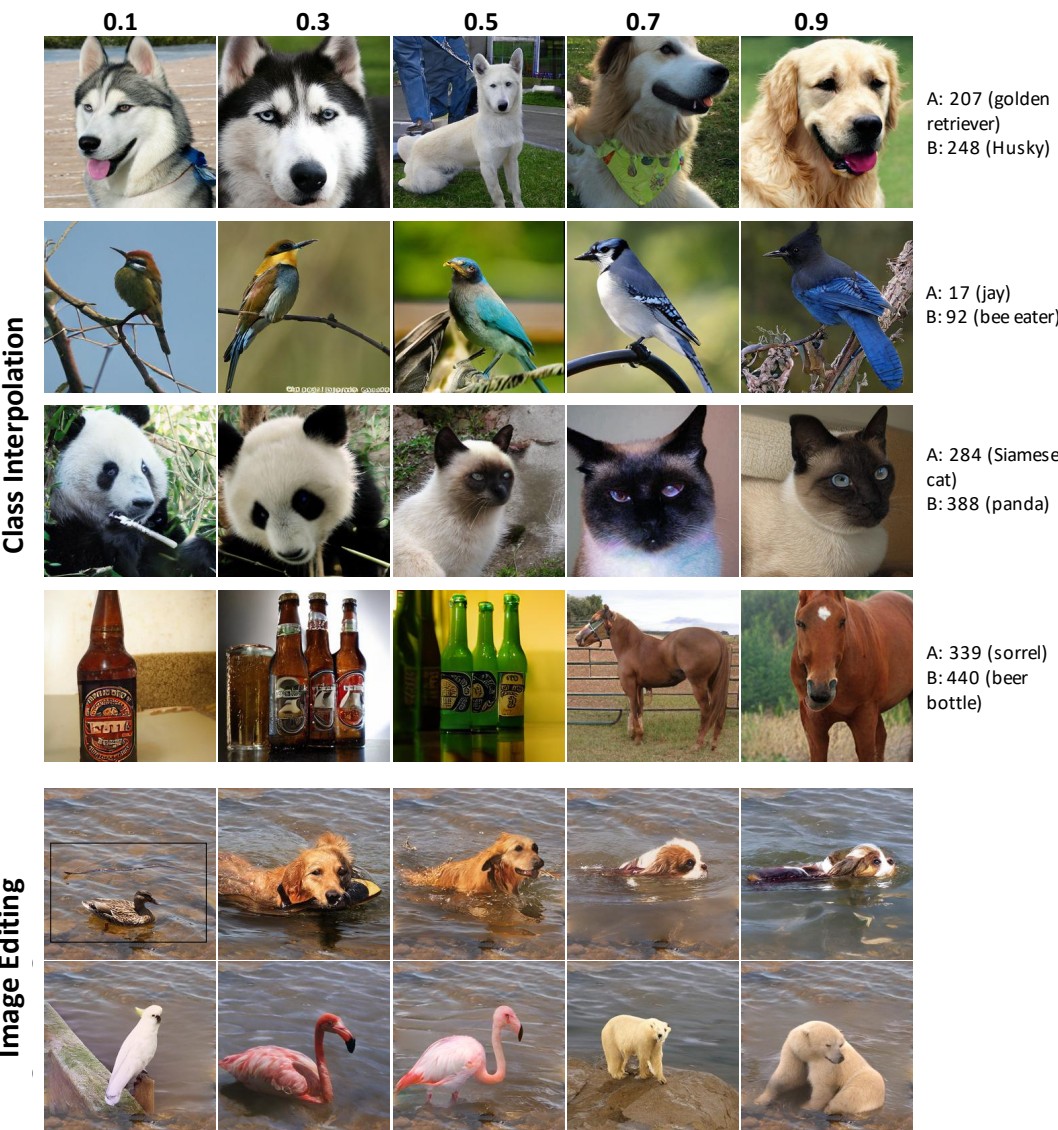

Figure 10: Zero-shot generalization performance of ELM. Class interpolation generate images with interpolated class condition, i.e., $\alpha A + (1-\alpha)B, \alpha \in \{0.1, 0.3, 0.5, 0.7, 0.9\}$. Image editing allows the model to edit the masked region based on specific class condition.

### A.3. Performance on different datasets

We also conduct experiments on specialized datasets distinct from ImageNet to assess the robustness and versatility of ELM models. Specifically, we select CelebA (Liu et al., 2015), which includes 202,599 human face images across 10,177 identities, and the Describable Texture Dataset (DTD) (Cimpoi et al., 2014) that comprises 5,640 images across 47 different categories. We train an ELM-L model with a 2-8 tokenizer on each dataset for 400 epochs using a batch size of 256. The qualitative results (**Figure** 11 and 12) from these experiments demonstrate the high performance of our model across diverse types of tasks.

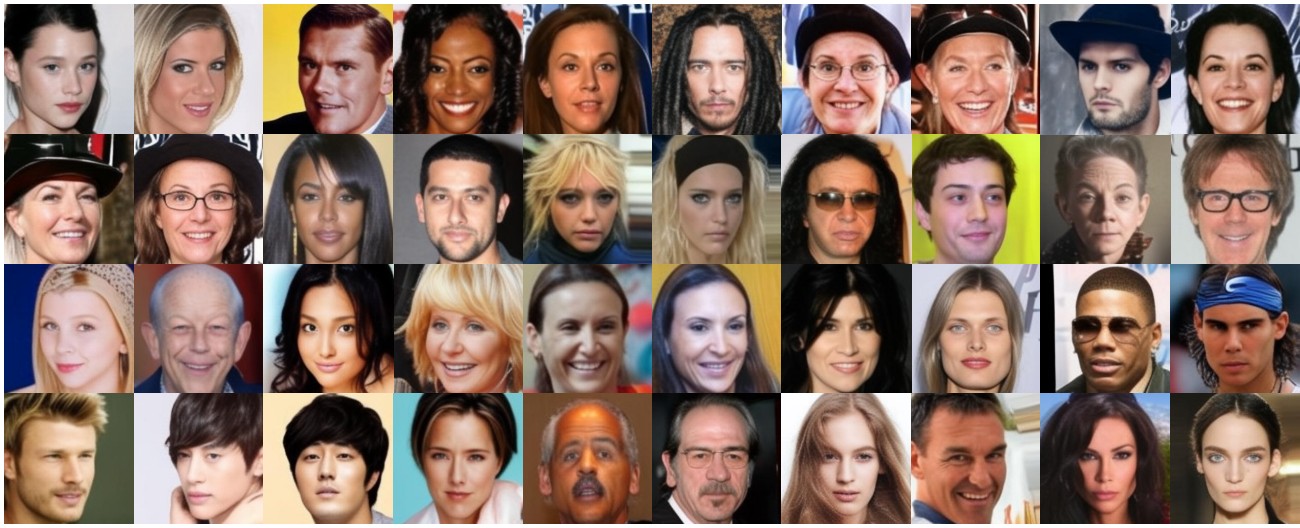

Figure 11: Generated human face images with 256×256.

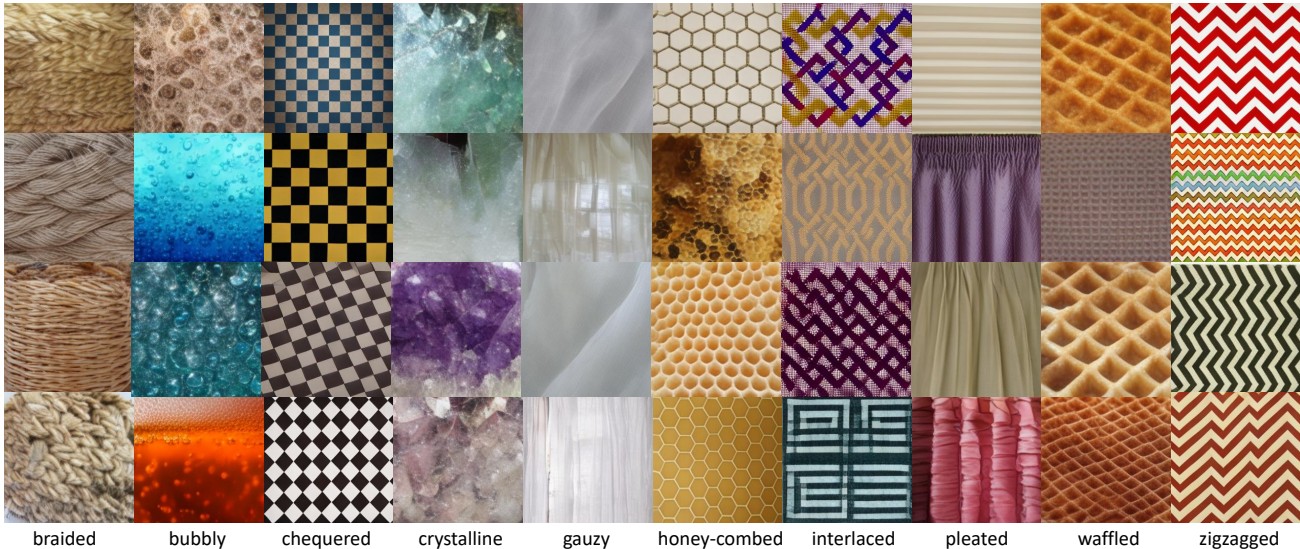

braided    bubbly    chequered    crystalline    gauzy    honey-combed    interlaced    pleated    waffled    zigzagged

Figure 12: Generated special texture images with 256×256.

### A.4. Intrinsic difference between language and images

We choose ImageNet from the image domain; OpenWebText and Shakespeare[2] from the language. The information of the tokenized dataset is shown in **Table** 5 and the KL-divergence between uniform distribution is shown in **Table** 6.

We can see that from **Table** 6, compared to text generation, image generation exhibits a higher randomness. Note that although VQGAN generates tokens with a lower level of randomness, the major reason is the low code utilization-only less than 10% code from the vocabulary is used, and the generated image quality is not satisfying due to the extremely low token utilization.

Table 5: Vocabulary (Codebook) information of image and text.

|  | **ImageNet** | | **OpenWebText** | **WallStreetJournal**[3] |
|---|---|---|---|---|
| Tokenizer | VQGAN | BAE | BPE | BPE |
| Vocab size | 16384 | 65536 | 47589 | 19979 |
| Token num of train set | 327M | 327M | 9B | 38M |
| Token num of val set | 12M | 12M | 4M | 1.5M |

Table 6: KL-divergence between token distribution and Uniform Distribution, along with the perplexity of n-gram models.

|  | **ImageNet** | | | | **OpenWebText** | | **WallStreetJournal** | |
|---|---|---|---|---|---|---|---|---|
| Tokenizer | VQGAN (V=16384) | | BAE (V=65536) | | BPE (V=47589) | | BPE (V=19979) | |
|  | unigram | bigram | unigram | bigram | unigram | bigram | unigram | bigram |
| Train | 1.00 | 2.16 | 0.24 | 0.17 | 3.25 | 3.35 | - | - |
| Val | 0.90 | 2.12 | 0.22 | 0.03 | 3.27 | 1.94 | - | - |
| Perplexity [4] | 368 | 210 [5] | 52,538 | 596,855 | 2087 | 395 | 962 | 170 |

### A.5. Additional Experiment Results

**Implementation Details** For the BAE tokenizer, we followed the configuration in (Wang et al., 2023), utilizing Bernoulli sampling during quantization, and trained it for 400 epochs on the ImageNet dataset. For the transformer model, we adopted the LLaMA-2 (Touvron et al., 2023b) architecture, as referenced in (Sun et al., 2024). The depth and feature dimensions of each model size are detailed in **Table** 7. All language models were trained on 80GB A800 GPUs with a batch size of 256, for 400 epochs, using a constant learning rate of 1e-4, weight decay of 0.05, and the AdamW optimizer with $\beta_1$ 0.9 and $\beta_2$ 0.95. The L and XL-sized models were trained on 8 A800 GPUs, requiring approximately 6.4 and 10 days, respectively, to complete 400 epochs. The XXL-sized model, trained on 16 A800 GPUs (2 nodes with 8 GPUs each), took around 12 days to finish training.

For the AR model, we implement mainly follow (Sun et al., 2024), except for the 2B-sized model. The MLM and AR models use the same model architecture. For the MLMs training strategy, we mainly follow (Chang et al., 2022). Specifically, at each training step, we sample a mask ratio for each sample, mask tokens based on this ratio, and train the model to predict the masked tokens. The mask ratio follows a cosine schedule across the generation iterations, meaning the process transitions from less to more information. Early in training, most tokens are masked; as training progresses, the mask ratio sharply decreases, revealing more tokens for the model to handle in later stages.

During the exploration of the *design space and other ablation studies*, we calculate the FID score using only **30,000** samples for efficiency, except for key comparisons such as language modeling choices (AR vs. MLM). Specifically, unless otherwise noted, the FID results in **Section** 3, where we elucidate the design space, are calculated based on 30,000 generated samples.

---

[2]Obtained from https://github.com/karpathy/nanoGPT

[3]The information is obtained from Standford lecture note: https://web.stanford.edu/ jurafsky/slp3/3.pdf

[4]We calculate the perplexity with Laplace smoothing(Gale & Church, 1994). The first 10 percent of the training data is select the efficiently calculate the perplexity of OpenWebText.

[5]Although the VQGAN tokenizer exhibits lower perplexity compared to BAE, its extremely low code utilization significantly impacting the tokenizer's effectiveness.

It is worth noting that the FID values obtained this way are consistently higher than those calculated with 50k samples. For a *fair* comparison with other methods, we generate **50,000** samples in the comparison results presented in the **Section** 4.

Table 7: Transformer model architecture information with different sizes.

| Size | depth | dimension | num of head |
|---|---|---|---|
| ELM-L | 24 | 1024 | 16 |
| ELM-XL | 36 | 1280 | 20 |
| ELM-XXL | 48 | 1536 | 24 |
| ELM-2B | 48 | 1792 | 28 |

### A.5.1. EFFICIENCY ANALYSIS

While our study primarily focuses on understanding the modeling behavior and generation quality of LLMs for image synthesis, we briefly summarize efficiency-related insights for completeness. Given that modern LLM acceleration techniques—such as KV-cache, FlashAttention, and quantization—are mature and directly transferable to our setting, we do not emphasize efficiency optimization in the main body.

Overall, AR models offer superior training efficiency, MLM achieves the fastest inference, and memory usage differences mainly emerge at inference due to AR's KV-cache overhead. Detailed comparisons are reported in **Table** 8.

Table 8: Comparison of training and inference efficiency across models.

| Model | Params (M) | FLOPs (G) | Training Epochs (Converge/Total) | Inference Time (s/img) |
|---|---|---|---|---|
| DiT/XL-2 | 675 | 118.64 | ∼160 / 1400 | 0.39 (50 steps) |
| MLM-XL | 741 | 189.51 | ∼200 / 400 | 0.10 (10 steps) |
| AR-XL | 741 | 189.98 | ∼100 / 400 | 0.15 (256 steps w/ KV-cache) |

### A.5.2. RESULTS OF DIFFERENT TOKENIZERS: BAE *v.s.* VQGAN

We trained BAE on the ImageNet dataset using the same model architecture and loss functions as VQGAN from the taming transformers framework (Esser et al., 2021). For a fair comparison, we evaluated the VQGAN-16384 model[6] that also trained on the ImageNet dataset, and assessed its code utilization (see **Figure** 13. The results clearly demonstrate that BAE outperforms VQGAN, achieving lower reconstruction FID (rFID) and generation FID (gFID) (**Table** 1 and 9) and significantly higher code utilization (100% v.s. 8%).

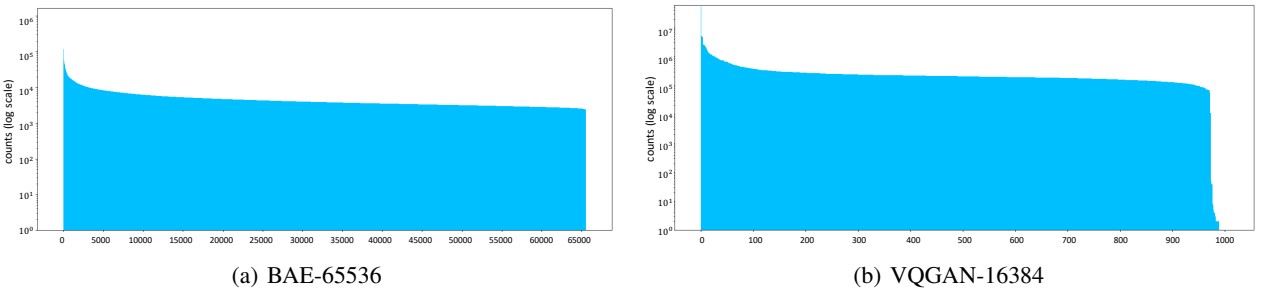

(a) BAE-65536        (b) VQGAN-16384

Figure 13: **BAE-16 exhibits a higher code utilization than VQGAN.** This figure shows a log count number of the appearance of codes on the ImageNet training dataset in sorted order. (a) BAE-16, with a code dimension of 16, has 65,536 unique codes and achieves 100% code utilization, with no code showing extremely low usage. In contrast, (b) VQGAN, with a codebook size of 16,384, only utilizes around 1,000 codes, and many of these codes have extremely low utilization.

---

[6]Downloaded from https://github.com/CompVis/taming-transformers

Table 9: **Generation FID of L models with different image tokenizers.** A. denotes AR, M. denots MLM. All models are trained on the ImageNet for 1,000,000 iterations, about 200 epochs. 'cfg1-3' denotes classifier-free guidance (cfg) scale gradually increased to 3.0 following a linear schedule across inference iteration. 'cfg1.5' denotes the cfg remains fixed at 1.5 during inference.

| tokenizer | code dim | v. size & top-$k$ | cfg1-3 FID(A.) | cfg1-3 FID(M.) | cfg1.5 FID(A.) | cfg1.5 FID(M.) |
|-----------|----------|-------------------|----------------|----------------|----------------|----------------|
| **VQGAN** | 256 | 16,384 | 6.71 | 7.81 | 8.12 | 8.98 |
| **BAE** | 16 | 65,536 | 2.78 | 3.96 | 3.87 | 5.01 |

Table 10: **The influence of Bernoulli sampling with BAE on FID (30k) of generation.** We test on AR-L model with BAE with $D = 16$, and the model is trained for 150 epochs.

| cfg | constant 2 | linear1-3 |
|-----|------------|-----------|
| w. Bernoulli | 4.72 | 2.88 |
| w.o. Bernoulli | 5.05 | 3.13 |

### A.5.3. RESULT OF BAE W. AND W.O BERNOULLI SAMPLING

When using BAE to tokenize image feature codes into discrete tokens, the process can either be deterministic, by directly converting values to 0 or 1 based on a threshold, or nondeterministic by incorporating Bernoulli sampling during quantization. We compared both methods to assess their impact on the generation task. As shown in **Table** 10, the nondeterministic approach clearly performs better. This result aligns with the inherent randomness of image token distribution, as discussed in **Section** 6, and offers greater tolerance for classification errors during next-token prediction.

### A.5.4. RESULTS OF DIFFERENT MODELING METHODS: AR *v.s.* MLM

**Table** 11 shows the detailed final result of the different-sized AR models and MLMs using the basic BAE on the ImageNet 256×256 dataset. Clearly, AR models always show better performance than MLMs.

Table 11: **Comparison of AR and MLM on ImageNet 256×256**. The auto-encoder is BAE with code dimension 16. The FID results are obtained on 50,000 generation images.

| Size | Method | FID↓ | sFID↓ | IS↑ | Precision↑ | Recall↑ |
|------|--------|------|-------|-----|------------|---------|
| L | MLM | 3.67 | 5.34 | 272.23 | 0.8561 | 0.4597 |
| | AR | 2.38 | 4.78 | 271.54 | 0.8201 | 0.5650 |
| XL | MLM | 3.13 | 4.95 | 261.59 | 0.8159 | 0.5355 |
| | AR | 2.14 | 4.92 | 289.33 | 0.8162 | 0.5834 |
| XXL | MLM | 3.12 | 4.86 | 281.75 | 0.8393 | 0.4947 |
| | AR | 2.10 | 4.89 | 301.22 | 0.8284 | 0.5839 |

### A.5.5. COMPARISON BETWEEN CODE-DECOMPOSITION STRATEGIES

**Table** 12 shows the detailed results of AR models with different BAE tokenizers. The code decomposition strategy significantly influences the model parameter size and the generation performance. For the code decomposition strategy, splitting a large vocabulary into **two** smaller sub-vocabularies yields optimal performance by *balancing vocabulary size with the number of classification heads*. However, using more than two classification heads increases the model's training complexity and impacts learning effectiveness. **Figure** 14 shows the training loss curves of AR models with 2-8 and 3-8 tokenizers. Both begin with the same initial loss, but clearly, models with 2-8 tokenizers converge better than those with 3-8 across various model sizes. This demonstrates that making three classifications at each position during next-token prediction significantly increases the learning complexity, affecting convergence efficiency.

In general, larger code dimensions improve generation performance by offering finer granularity, but they also introduce more complex vocabularies, making it increasingly challenging for the model to predict the next token accurately, therefore,

the vocabulary size should match the generation model's capacity.

Table 12: **Comparisons of AR models on class-conditional ImageNet 256×256 benchmark.**

| Model | Tokenizer | Params. | FID↓ | sFID↓ | IS↑ | Precision↑ | Recall↑ |
|-------|-----------|---------|------|-------|-----|------------|---------|
| L | 1-16 | 443M | 2.38 | **4.78** | 271.54 | 0.8201 | 0.565 |
| | 2-8 | 312M | 2.34 | 4.86 | 281.29 | 0.8190 | 0.5573 |
| | 2-10 | 316M | **2.17** | 4.83 | 288.59 | 0.8168 | 0.5536 |
| | 2-12 | 328M | 2.34 | 5.12 | **316.08** | 0.8197 | 0.5487 |
| XL | 1-16 | 900M | 2.14 | 4.92 | 289.33 | 0.8162 | 0.5834 |
| | 2-8 | 737M | 2.01 | 4.50 | 298.99 | 0.8069 | 0.5979 |
| | 2-10 | 741M | **1.73** | **4.50** | **332.38** | 0.8183 | 0.5823 |
| | 2-12 | 757M | 1.79 | 4.82 | 328.99 | 0.8027 | 0.5903 |
| | 3-8 | 740M | 1.99 | 5.29 | 329.66 | 0.8070 | 0.5906 |
| XXL | 1-16 | 1.56B | 2.10 | 4.89 | 301.22 | 0.8284 | 0.5839 |
| | 2-10 | 1.37B | 1.65 | **4.33** | 328.08 | 0.8144 | 0.5933 |
| | 2-12 | 1.39B | **1.58** | 4.78 | **330.43** | 0.8034 | 0.6091 |
| | 3-8 | 1.37B | 1.67 | 4.99 | 325.06 | 0.8020 | 0.6054 |
| | 4-8 | 1.37B | 2.02 | 5.66 | 321.37 | 0.7913 | 0.602 |
| 2B | 2-12 | 1.90B | **1.54** | **4.81** | **332.69** | 0.8093 | 0.5968 |

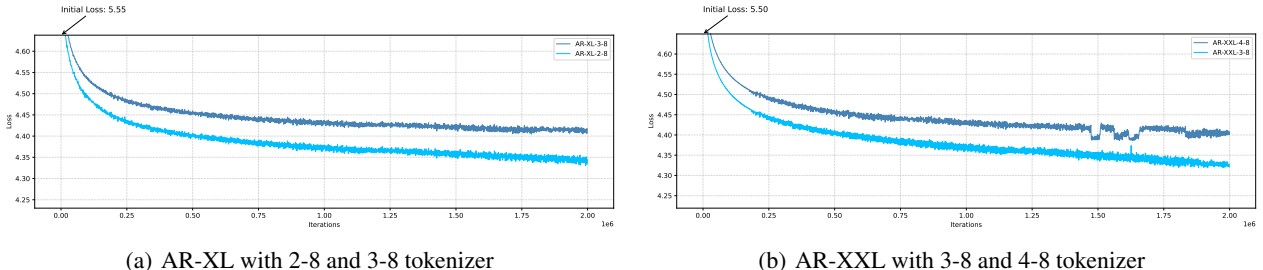

(a) AR-XL with 2-8 and 3-8 tokenizer          (b) AR-XXL with 3-8 and 4-8 tokenizer

Figure 14: **Different vocabulary decomposition strategies vary a lot on the training losses.** Clearly, introducing more than two classification heads will increase the model's training complexity and learning effectiveness.

### A.5.6. SCALING BEHAVIOR OF AR MODELS

To further illustrate the scaling behavior of AR models, in addition to the learning pattern visualization and image generation results we present on the main pages, we also provide the convergence curves during training. **Figure** 15 shows the loss trends for all AR model sizes (L, XL, XXL, and 2B) with the 2-12 tokenizer. All models successfully converged, and the final loss consistently decreased as the model size increased.

### A.5.7. COMPARISON BETWEEN SAMPLING STRATEGIES

For MLMs, we conduct a search to find the optimal CFG scale, iteration number, and temperature $\tau$ for the Gumbel noise. For AR models, we search for the best CFG scale and top-$k$ threshold.

Classifier-free guidance (CFG) plays a crucial role in conditional image generation, but it involves balancing the trade-off between image diversity and individual image quality. We searched for the optimal CFG scale for all models. Additionally, we found that using a dynamic CFG schedule significantly improves performance. We tested several CFG scheduling methods (see **Figure** 16), with the results summarized in **Table** 13. **Figure** 17 and 18 further show the qualitative comparison of different CFG scale.

Notably, the effect of top-$k$ varies substantially between vision and language domains. In image generation, small $k$ (e.g.,

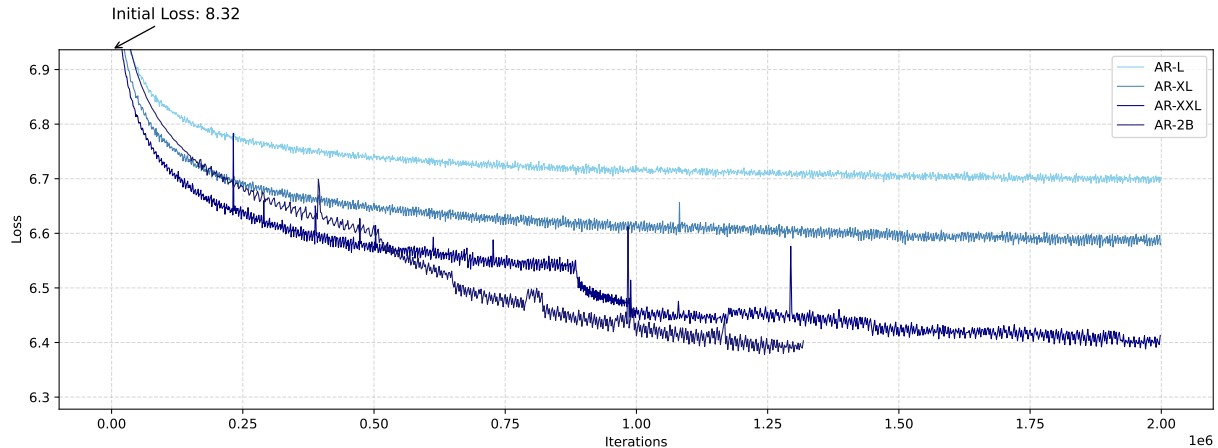

Figure 15: **AR exhibits a good scaling law.** Training losses of all AR models are with the BAE 2-12 tokenizer. All models were trained for 2,000,000 iterations, equivalent to 400 epochs, except the 2B model, which had to be stopped earlier due to time constraints.

Table 13: **Different CFG strategies vary greatly on FID.** All results are based on AR-L with tokenizer 2-10.

| CFG-scale | 1.5 | 2 | 2.5 | cos1-4 | log1-4 | linear1-4 | square1-4 | r-square1-4 |
|---|---|---|---|---|---|---|---|---|
| FID (30k) | 2.98 | 3.35 | 3.58 | 2.86 | 2.70 | **2.48** | 4.94 | 3.57 |

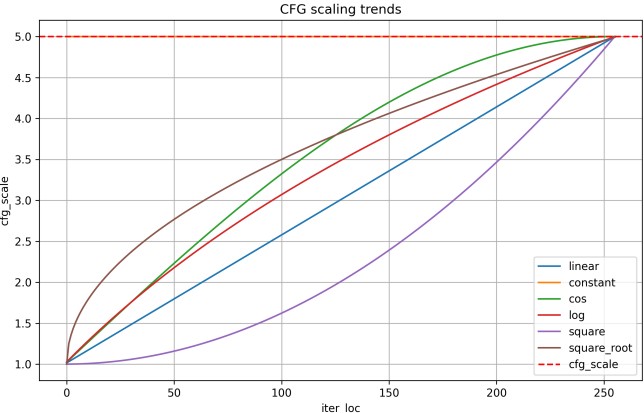

Figure 16: **Curves of CGF scale with respect to iteration times under different CFG schedules.**

Table 14: **The influence of top-$k$ in sampling process** for AR models with decomposed vocabulary.

| | 2-8 | | | 2-10 | | | 2-12 | | |
|---|---|---|---|---|---|---|---|---|---|
| $k$ | 180 | 210 | 256 | 800 | 900 | 1024 | 2600 | 2800 | 3000 |
| L | 2.97 | 2.84 | **2.74** | 2.55 | 2.50 | **2.48** | 2.68 | **2.56** | 2.67 |
| XL | 2.46 | **2.36** | 2.40 | 2.13 | 2.11 | **2.03** | 2.11 | **2.10** | 2.11 |
| XXL | - | - | - | 2.08 | 2.04 | **1.95** | **1.90** | **1.90** | 1.95 |

<100) leads to overly smooth, low-diversity outputs, and optimal values often reach $k \approx 0.5\times$ vocabulary size to capture texture richness (**Figure** 19). In contrast, text generation typically requires only $k = 20$–$100$ to achieve diverse, fluent

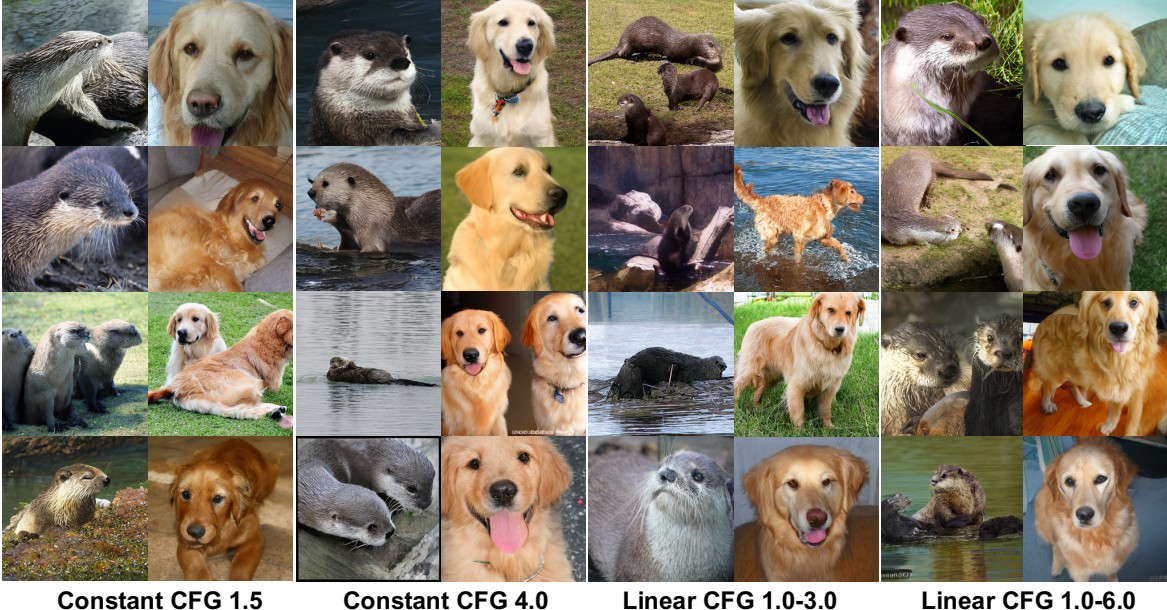

| Constant CFG 1.5 | Constant CFG 4.0 | Linear CFG 1.0-3.0 | Linear CFG 1.0-6.0 |

Figure 17: **Visual comparison of different CFG setting. The top-$k$ is set to 500.** Clearly, compared to a constant CFG value, a gradually increasing CFG value tends to enhance image diversity and the alignment with the real data distribution, explains a better FID score. Besides, smaller CFG value also brings more diversity and more background details.

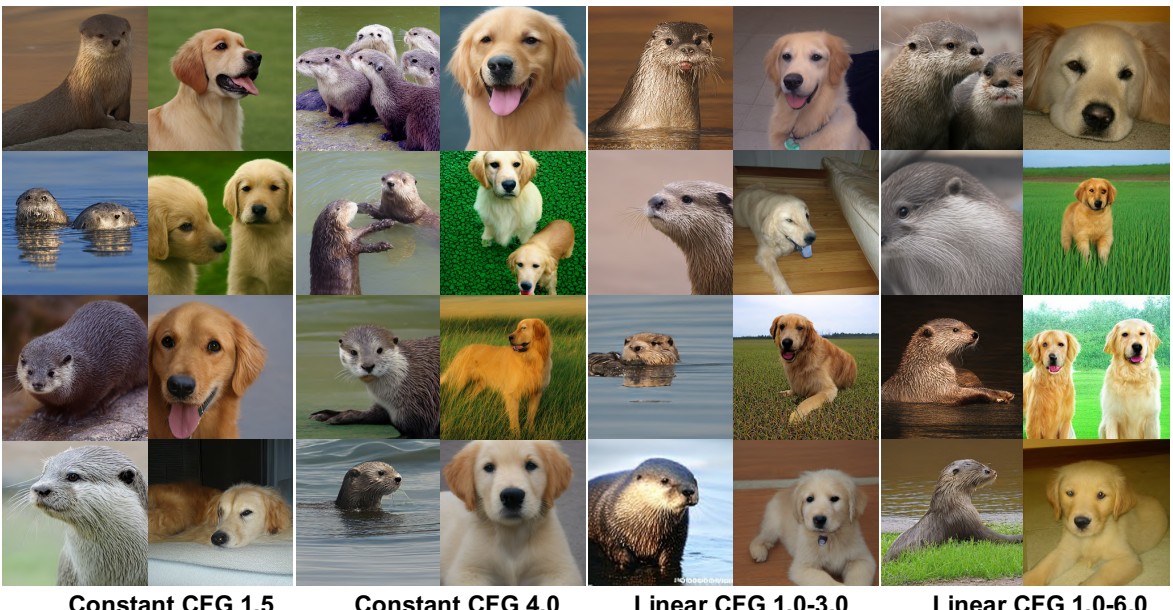

| Constant CFG 1.5 | Constant CFG 4.0 | Linear CFG 1.0-3.0 | Linear CFG 1.0-6.0 |

Figure 18: **Visual comparison of different CFG settings. The top-$k$ is set to 100**. Compared to the CFG scale, top-$k$ setting has more influence on the image visual quality. When $k$ is small, a linearly increased CFG scale will bring more diversity and background information.

outputs, even with vocabularies of 50,000–128,000 tokens (Holtzman et al., 2019; Fan et al., 2018). This reflects fundamental differences in the distributional properties of visual and linguistic data.

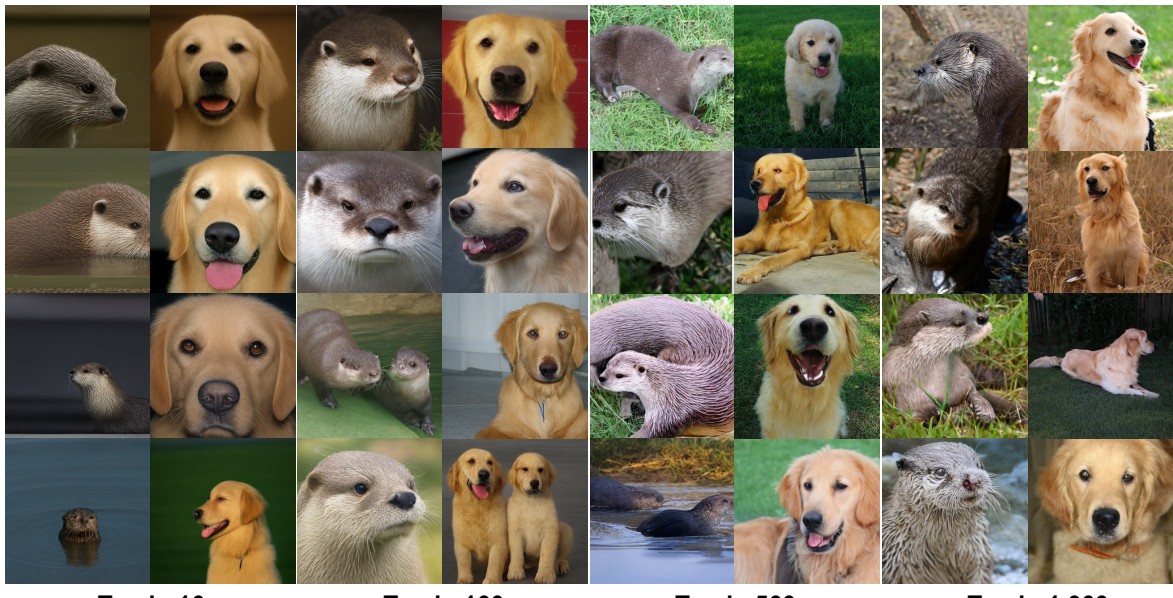

**Top-k: 10**  **Top-k: 100**  **Top-k: 500**  **Top-k: 1,000**

Figure 19: **Visual comparison of different top-$k$ settings. The CFG is set to a constant value of 2.** Clearly, when the value of $k$ is small, the generated images tend to be overly smooth and sharp, lacking background and fine details. In contrast, larger $k$ values lead to images with richer details that better align with the distribution of real-world datasets.

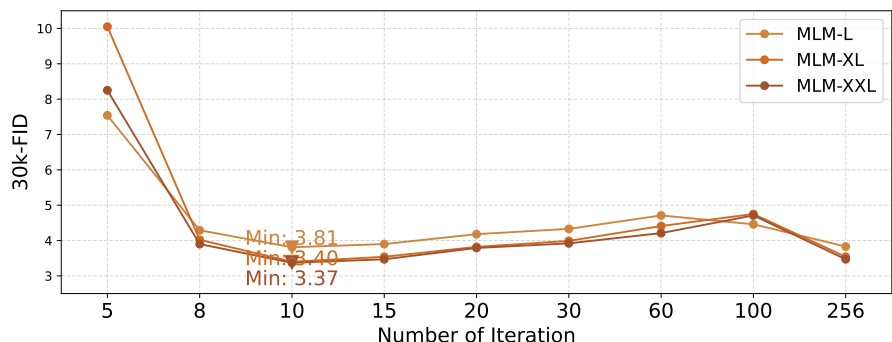

Figure 20: **The influence of iteration time (different mask ratio) in sampling process** on FID scores for MLMs.

Table 15: The best sampling strategy regard to FID score for all models.

| Method | Tokenizer | Model | Best Strategy |
|---|---|---|---|
| MLM | 1-16 | L | linear CFG 1-3; $\tau$=9.0, iteration number=10 |
| | 1-16 | XL & XXL | linear CFG 1-3; $\tau$=5.0, iteration number=10 |
| AR | 1-16 | L& XL & XXL | linear CFG 1-3; top-$k$=65536 (all) |
| | 2-8 | L | linear CFG 1-4; top-$k$=256 (all) |
| | 2-8 | XL & XXL | linear CFG 1-4; top-$k$=210 |
| | 2-10 | L | linear CFG 1-4; top-$k$=1024 (all) |
| | 2-10 | XL & XXL | linear CFG 1-5; top-$k$=1024 (all) |
| | 2-12 | L & XL & XXL | linear CFG 1-5; top-$k$=2800 |
| | 3-8 | XL & XXL | linear CFG 1-5; top-$k$=180 |
| | 4-8 | XXL | linear CFG 1-5; top-$k$=180 |

### A.6. The flexibility of ELM to generate any-size image

To further explore the capability of AR models in image generation, we generate images with more than 16×16 tokens without modifying the model (**Figure** 21). Although the model's receptive field is limited to 256 tokens, we can easily generate 'streaming' images by looking back at a few tokens. This demonstrates the greater flexibility of AR models compared to diffusion models, highlighting the potential of AR models for applications in other domains.

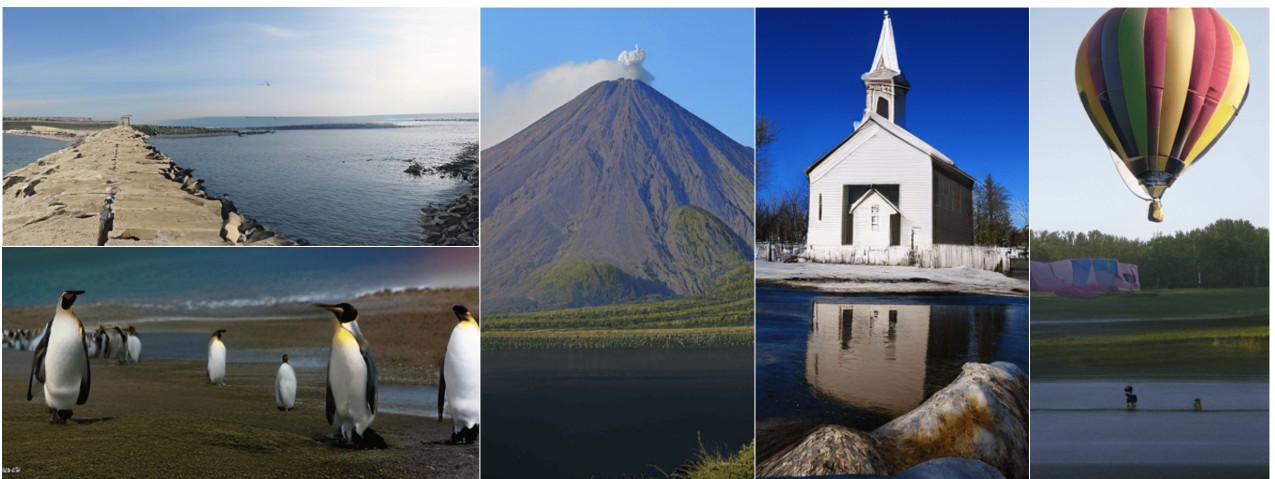

Figure 21: AR models are flexible to generate images with any size based on previous context.

### A.7. Limitation

Our work has limitations. While we propose several improvements for AR models, the fundamental issue of optimizing highly random token distributions remains. Traditional next-token prediction using classification loss may not be the most optimal training objective for such tasks, suggesting that more suitable objectives should be explored in future research. For instance, MAR (Li et al., 2024) has made promising progress by introducing diffusion loss into AR models, while VAR (Tian et al., 2024) presents a valuable perspective by altering the image tokenization approach. We hope our analysis will inspire further exploration and innovation in utilizing language models for vision generation, as well as other modalities.

### A.8. More generated samples

We present more randomly picked $256 \times 256$ samples here to straightforwardly show the performance of our model.

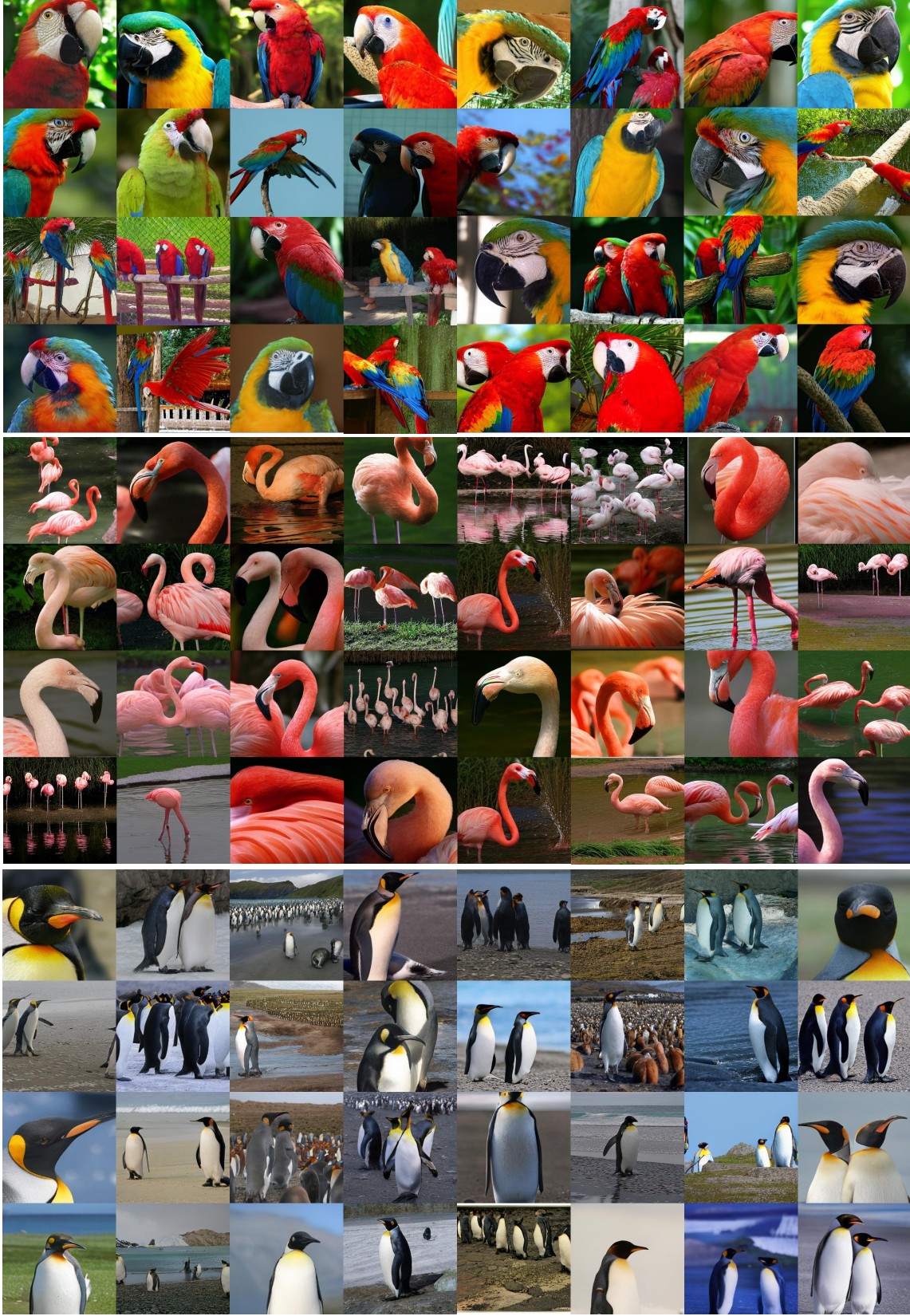

Figure 22: Randomly sampled images from classes 88 (macaw), 130 (flamingo), and 145 (king penguin).

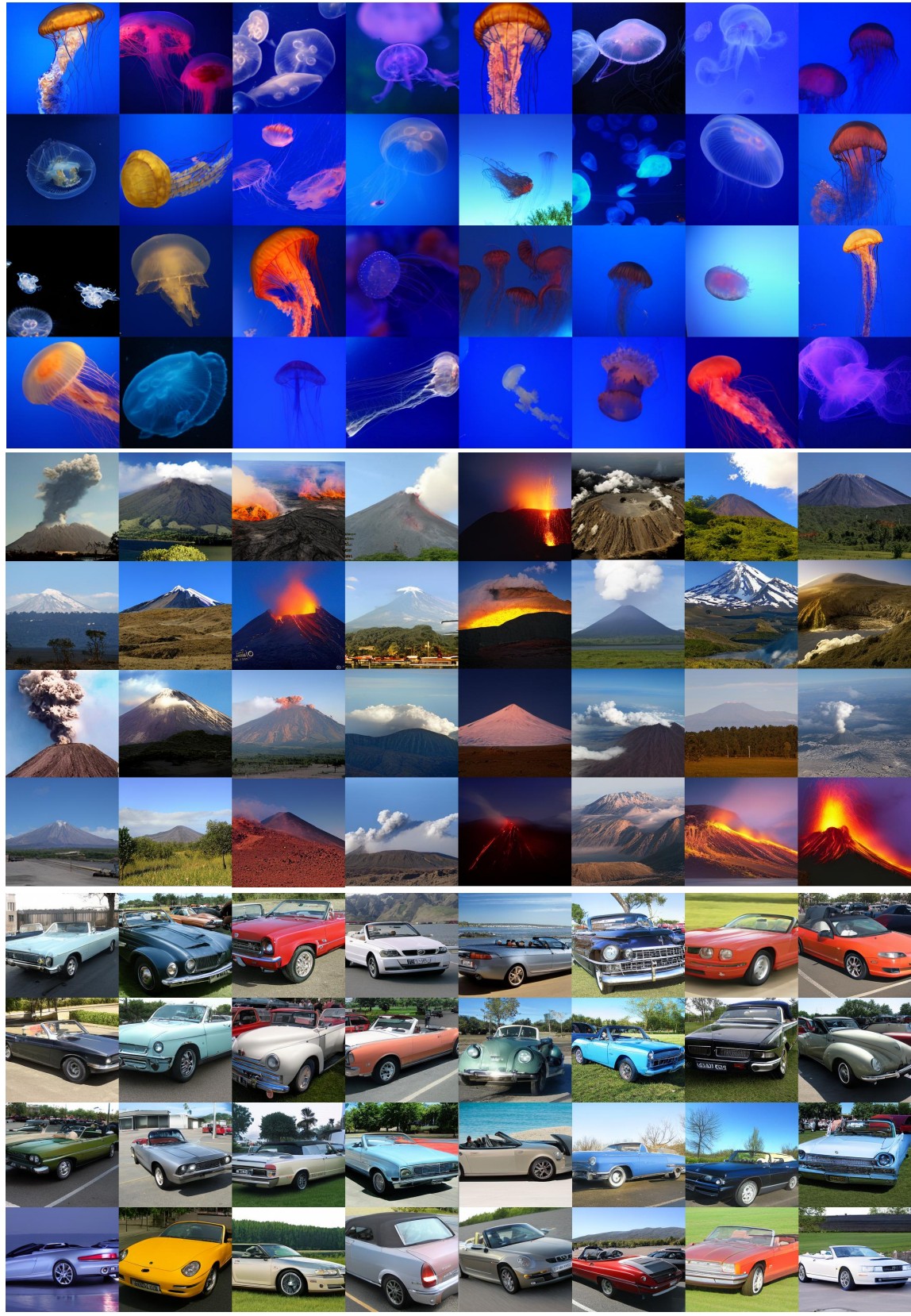

Figure 23: Randomly sampled images from classes 107 (jelly fish), 980 (volcano), and 511 (convertible).

