# OpenReview forum: "Elucidating the design space of language models for image generation"
_ICML.cc/2025/Conference — ICML 2025 poster_

### Official Review · Reviewer_uJCp · 2025-03-10

**Overall Recommendation:** 2

**Summary:**

The paper investigates the application of large language models (LLMs) to image generation, demonstrating that LLMs can achieve near state-of-the-art performance without relying on domain-specific designs. The study also analyzes the learning and scaling behavior of autoregressive models, showing that larger models capture more useful information.

**Claims And Evidence:**

The claims made in the submission are supported by clear and convincing evidence.

**Essential References Not Discussed:**

Two highly relevant papers should be discussed and made for fair comparison:

[1] Sun, Peize, et al. Autoregressive Model Beats Diffusion: Llama for Scalable Image Generation

[2] Li, Tianhong, et al. Autoregressive image generation without vector quantization

**Experimental Designs Or Analyses:**

The experimental designs are valid.

**Methods And Evaluation Criteria:**

The evaluation criteria generally make sense.

**Other Comments Or Suggestions:**

No

**Other Strengths And Weaknesses:**

Weaknesses
1. The technical novelty is kind of limited, as the results are mostly empirical.
2. This paper claims AR "demonstrate superior image generation ability and scalability compared to MLMs" in L089 and L090, however, works done to improve MLM for image generation in [1] shows much greater results. Authors should make fair comparison with [1] to defend this point.
3. More qualitative study and empirical discussion on cfg and top-k are important to understand how image generation differ from original decoding strategy in LLM.

[1] Li, Tianhong, et al. Autoregressive image generation without vector quantization

**Questions For Authors:**

I'm leaning towards weak reject before rebuttal. However, this research is in good shape with valid contributions. I would be happy to raise my score to 3 (weak accept) if my concerns in [W2] and [W3] and quantitative comparisons with [1][2] are addressed during rebuttal.

[1] Li, Tianhong, et al. Autoregressive image generation without vector quantization

[2] Sun, Peize, et al. Autoregressive Model Beats Diffusion: Llama for Scalable Image Generation

**Relation To Broader Scientific Literature:**

This paper shows that LLMs can achieve strong image generation performance without domain-specific designs, through careful choices in tokenization, modeling, and sampling. It contributes to understanding scaling behaviors and modality differences, offering insights for adapting LLMs to vision tasks and encouraging broader cross-domain applications.

**Theoretical Claims:**

The theroretical claims are correct

---

> ### Author Rebuttal · Authors · 2025-04-01
>
> Thanks a lot for your valuable comments. Below, we will address your concern in detail.
>
> ``` Discuss with LlamaGen and MAR```
>
> We agree that both [1]LlamaGen(Sun et al.) and [2]MAR(Li et al.) have made important contributions to advancing autoregressive image generation. We would like to clarify that we **do** compare against both works in **Table 1**, and further discuss MAR in the appendix (**A.7 Limitation section**) as a **parallel study** that proposes new modeling approaches and domain-specifc design for AR to effectively work in the **continuous** image domains.
>
> Compared to MAR, our work is more aligned with LlamaGen, as we both follow the same direction of **directly adapting language models for image generation** to explore their modeling potential. However, our work differs in that we conduct a **more comprehensive and systematic analysis** of the image-as-language modeling framework. In particular, we study:
> * The effect of tokenizer choice (VQGAN vs. BAE),
> * Modeling paradigms (AR vs. MLM),
> * Detailed learning behavior reflected by attention pattern and loss dynamics,
> * Vocabulary design through scalable codebooks under BAE,
> * And an in-depth ablation and analysis of sampling strategies.
>
> We believe our findings provide a **complementary perspective** to LlamaGen and MAR, and offer insights that could be valuable when combined with more advanced objectives or tokenization schemes in future work. We appreciate the suggestion and will incorporate the detailed discussions more clearly in the revised version.
>
> ```W1. technical novelty```
>
> We believe that our work makes a **meaningful and timely** contribution by **systematically analyzing** the design space of applying **language models in the image domain**, providing insights into adopting LLMs as a unified method and guiding future methodological innovations, especially in light of recent developments of GPT-4o, a powerful natively AR large multimodal model. You may find our detailed response to **W1 from Reviewer xmQD**.
>
> ```W2. Comparison with MAR```
>
> Thank you for pointing out that our conclusion on the choice of generation paradigm differs from MAR. However, given the differences in our methods, we believe this divergence is entirely reasonable.
> * We adopt a standard language modeling view, treating images as **discrete token sequences** with no image-specific modifications. Under this setup, we systematically compare AR and MLM using **standard tokenizers** and sampling, finding AR consistently superior in quality and scalability. Besides, this discrete token framework can integrate well with **LLM-based multimodal extensions**.
> * MAR presents a **new generative modeling approach** tailored for **continuous** image data with **diffusion loss** and **no quantization**. Their finding that MLM outperforms AR is valid within this setting but reflects a **fundamentally different modeling goal and assumptions** from ours.
>
> We view these works as complementary: MAR explores **domain-specific continuous modeling**, while we investigate how **standard language modeling paradigms transfer to vision**. Based on the different tokenizers and training objectives, we come to different conclusion according to the generation modeling method, which totally makes sense. We have acknowledged this in our appendix and will add the discussion to the main paper.
>
> ```W3. qualitative study and empirical discussion on top-k and cfg```
>
> We agree that understanding top-k sampling and classifier-free guidance (CFG) is key to characterizing image generation behavior, especially in contrast to standard LLM decoding. We have included an extensive analysis of both strategies (Figure 6, Table 12). We further provide **qualitative** comparisons (linked [here](https://anonymous.4open.science/r/ICML2025rebuttal-D02C/qualitive_topk_CFG.pdf)). We find that image generation requires **larger top-k values** and **moderate CFG weights** to achieve a better FID score. However, there is a **trade-off** between FID score and visual quality. A large k (close to the vocab size) introduces more texture details and diversity, making the outputs closer to the real distribution and thus yielding lower FID. In contrast, a moderate k often produces visually more appealing results, with cleaner and smoother images.
>
> While CFG is rarely applied in LLM-based text generation, top-k sampling is widely adopted in both domains:
> * In image generation, small k (less than 100) values lead to **overly smooth, low-diversity images**, lacking texture richness. Conversely, image tokens require broader sampling (e.g., **k ≈ 0.5 * vocab size**) to maintain realism.
> * In language generation, a typical k lies in the **20–100** range with a vocab size of **~50,000 or 128,000**, and can produce **diverse** and **grammatical fluency** texts (Holtzman et al., 2020; Fan et al., 2018).
>
> ref: Holtzman et al, 2020, The Curious Case of Neural Text Degeneration.
>
> Fan et al, 2018, Hierarchical Neural Story Generation.

---

> > ### Comment · Reviewer_uJCp · 2025-04-07
> >
> > I appreciate the authors’ detailed responses, which have addressed most of my concerns about the paper. The additional experiments on top-k and CFG are quite impressive.
> >
> > However, after carefully reading the other reviewers' comments and the authors’ rebuttal, I noticed that a substantial number of new experiments and results were introduced in the rebuttal, many of which are important to the overall claims of the paper. Given the empirical nature of the work and the limited technical novelty, I believe these new results should be properly integrated into the main paper and go through a full round of peer review rather than being evaluated solely during the rebuttal phase.
> >
> > While I acknowledge the consensus among the other reviewers and the authors’ considerable effort, I am therefore leaning slightly toward a weak reject this time. That said, I would still be OK if paper is ultimately accepted.

---

> > > ### Author Response · Authors · 2025-04-07
> > >
> > > Thank you for your thoughtful feedback and for recognizing the value of our work and rebuttal efforts.
> > > We’d like to clarify that the additional results provided during the rebuttal mainly include:
> > >  * comparisons between tokenizers (added VQGAN from LlamaGen) [for Reviewer ED5q]
> > >  * extended ImageNet-512 results (added VAR result) [for Reviewer xmQD]
> > >  * computational efficiency analysis (training/inference/memory) [for Reviewer e5YQ],
> > >  * qualitative results on sampling strategies (top-k and CFG).
> > >
> > > These additions help strengthen and clarify our statements—such as (1) demonstrating that bit-wise quantization (BAE) is more effective than vector-wise quantization (VQGAN), (2) our AR model with BAE can scale efficiently to high-resolution image generation, (3) AR models are not only effective but also competitive in overall efficiency, and (4) sampling strategies (e.g., top-k, CFG) have a significant impact on LM-based image generation—which we have included in the *Introduction* and *Experiment section* in our submitted paper. These are **complementary results** that extend and reinforce our existing claims **without altering** the main conclusions or methodology. We will carefully incorporate these findings into the revised version for completeness.
> > >
> > > We believe our study provides **timely and meaningful** insights for the community, especially toward understanding how AR modeling paradigms can serve as a unified foundation for multimodal generation and reasoning.

---

### Official Review · Reviewer_xmQD · 2025-03-13

**Overall Recommendation:** 3

**Summary:**

This paper systematically explores how to utilize LLMs for image generation, providing detailed comparisons and analyses across tokenization methods, modeling approaches, scan patterns, vocabulary design, and sampling strategies, offering some interesting conclusions. Based on these integrated experiments and analyses, the authors propose Elucidated Language Models, achieving remarkably good performance on ImageNet class conditional generation, demonstrating the significant potential of using LLMs for image generation.

**Claims And Evidence:**

Yes

**Essential References Not Discussed:**

NA

**Experimental Designs Or Analyses:**

Yes. I checked the results of the proposed method on class-conditional generation for ImageNet-256 and ImageNet-512, as well as the ablation studies in the paper.

**Methods And Evaluation Criteria:**

Yes

**Other Comments Or Suggestions:**

NA

**Other Strengths And Weaknesses:**

## Strengths
1. This paper thoroughly explores and analyzes the impact of different designs on LLM performance for image generation, providing valuable analysis and experiments for LLM-based Image Generation.
2. Based on these designs, the authors propose a strong baseline, achieving competitive results on ImageNet-256 and ImageNet-512 class-conditional image generation benchmarks.
3. The paper is written clearly and is easy to understand.

## Weaknesses
1. The paper doesn't offer much methodological innovation, but rather focuses on analyzing which existing designs are more suitable for LLM-based image generation.
2. There are relatively few baselines compared on the ImageNet-512 Benchmarks - what is the reason for this? VAR also has ImageNet-512 class-conditional image generation results, why weren't these compared?
3. The scaling law in Figure 7 only provides visual effects. Could the authors provide a scaling law for FID similar to the one in the VAR paper?

**Questions For Authors:**

NA

**Relation To Broader Scientific Literature:**

NA

**Theoretical Claims:**

There are no theoretical claims in this paper.

---

> ### Author Rebuttal · Authors · 2025-04-01
>
> Thanks for your valuable comment! Below, we will address your concern in detail.
>
> ```W1. methodological innovation```
>
> We acknowledge that our work does not center around proposing a new model architecture, but rather focuses on **systematically analyzing and understanding** how existing design components interact within the context of **LLM-based image generation**. We believe this perspective is **important and timely**, especially in light of recent developments such as **GPT-4o**, which has been introduced as a **natively AR large multimodal model** capable of cross-modal reasoning and generation, demonstrating impressive performance across tasks. These advances further support the need for a deeper understanding of AR modeling as a unified generative paradigm, and our study lays the **empirical foundation** for such future multimodal research. In particular, our contributions lie in:
> * Providing a fair comparison between AR and MLM under standard image tokenization;
> * Demonstrating how tokenization and dictionary structure affects model performance and interacts with model scaling;
> * Revealing surprising robustness of AR LMs to image token randomness and their ability to generate high-fidelity outputs;
> * Offering design principles that can guide future LLM-based multimodal generation systems.
>
> We view this paper as an **analytical and foundational contribution**, aiming to guide future methodological innovations. We appreciate the reviewer’s suggestion and will clarify this positioning in the revised version.
>
> ```W2. baselines compared on the ImageNet-512```
>
> On the ImageNet-512 benchmark, we did **not** re-train a model from scratch at 512×512 resolution. Instead, we performed a **lightweight fine-tuning** of our model that was originally trained on 256×256 images, using only **a few** additional training steps. Our primary goal in this experiment is to demonstrate the **resolution scalability and efficiency** of ELM, so we only choose some typical methods to compare with. Training directly on 512-resolution images from scratch would require approximately **4× more compute** compared to 256-resolution training. However, with a token-prediction-based architecture, we find that simple fine-tuning can achieve strong high-resolution performance with significantly reduced cost—a major advantage over diffusion models, which typically require retraining and redesigning the entire pipeline for higher resolutions.
>
> As a reference baseline, we chose to compare against DiT, a strong class-conditional diffusion model trained natively on 512×512 images. We also included MaskGIT as a discrete-token-based generation model for completeness. We acknowledge that VAR has also reported ImageNet-512 results. Below, we include a comparison with VAR.
> | Model | Params | Steps | FID ↓ | IS ↑ | Precision ↑ | Recall ↑ |
> |-------|---------|--------|--------|--------|-------------|-----------|
> | DiT-XL/2 | 675M | 3000k | 3.04 | 240.82 | 0.84 | 0.54 |
> | MaskGIT | 227M | 1500k | 7.32 | 156.00 | 0.78 | 0.50 |
> | VAR-d36-s | ~2B | 1250k | **2.63** | 303.2 | - | - |
> | ELM-L (2-8) | 312M | 250k | 4.82 | 246.87 | 0.81 | 0.59 |
> | ELM-XL (2-12) | 757M | 250k | 3.29 | **321.21** | 0.81 | 0.60 |
>
> Although our results are slightly behind those of VAR, it is important to note that VAR uses more than 2× the number of parameters and was trained for a significantly longer time.
>
> ```W3. scaling law for FID```
>
> Thank you for the helpful suggestion. While Figure 7 primarily illustrates visual improvements across scales, we would like to clarify that we **do provide quantitative evidence** of a scaling law for FID throughout the paper:
> * In Table 1, we report FID scores across models with increasing capacities, showing a clear performance improvement as model size increases.
> * In Figure 2, we plot FID curves over training epochs for AR and MLM models of different sizes, which further demonstrates the impact of scaling on convergence and generation quality.
> * In the Appendix (Table 11), we present a comprehensive breakdown of FID scores for AR models across multiple scales—from L, XL, XXL to 2B—under different vocabulary designs, further reinforcing the consistency of the observed scaling trend.
>
> Taken together, these results clearly support a scaling law for FID in AR-based image generation under the image-as-language paradigm, similar in spirit to what is shown in the VAR paper. We will make this connection more explicit in the revised version, and thank you again for encouraging us to clarify this aspect.

---

> > ### Comment · Reviewer_xmQD · 2025-04-03
> >
> > Thanks for the rebutall, my questions are well resolved by the authors. Hence, I will keep my original rating.

---

> > > ### Author Response · Authors · 2025-04-07
> > >
> > > Thank you for taking the time to review our rebuttal and for your thoughtful evaluation. We appreciate your assessment and support.

---

### Official Review · Reviewer_e5YQ · 2025-03-13

**Overall Recommendation:** 3

**Summary:**

This paper systematically explores the design space of large language models for image generation, evaluating factors such as tokenization, model architecture, scanning patterns, vocabulary decomposition, and sampling strategies. The proposed ELM achieves state-of-the-art FID scores, demonstrating the potential of LLMs for vision tasks. Key findings include: 1. Autoregressive models outperform masked language models for image generation, benefiting from sequential modeling. 2. Binary autoencoder tokenization is superior to VQGAN, reducing codebook collapse and improving reconstruction quality. 3. Row-major raster scanning yields the best performance among different image tokenization strategies. 4. Scaling laws hold for AR-based image generation, with larger models capturing both local and global features. Despite promising results, the study lacks comparisons with the latest diffusion models, does not evaluate text-to-image generation, and does not analyze computational efficiency, leaving room for further investigation.

**Claims And Evidence:**

Supported Claims:
- The paper claims that large language models (LLMs) can achieve near state-of-the-art image generation without specialized inductive biases, by optimizing tokenization, modeling, scanning patterns, vocabulary, and sampling strategies.
- Experimental results support this claim, as the proposed ELM model achieves an FID of 1.54 (256×256) and 3.29 (512×512) on ImageNet, which is comparable to leading diffusion-based methods.
- The study asserts that autoregressive (AR) models outperform masked language models (MLMs) in image generation, which is well-supported by quantitative comparisons (lower FID, better scalability).

Potentially Problematic Claims:
- The assumption that better tokenization alone can lead to improved performance is somewhat oversimplified—while BAE performs better than VQGAN, the impact of other factors such as model architecture and training strategy is not isolated/detailed in ablations.

**Essential References Not Discussed:**

None

**Experimental Designs Or Analyses:**

Strengths:
- Comprehensive exploration of design choices: The paper systematically evaluates tokenization, modeling, scanning patterns, vocabulary decomposition, and sampling strategies.
- Comparative performance analysis: The study includes quantitative results on multiple LLM architectures and different vocabularies, providing strong empirical support for its claims.
- Visualization of scaling laws: The analysis of attention patterns in AR models provides useful insights into how larger models learn hierarchical image representations.

Weaknesses:
- No efficiency analysis: The study does not compare training/inference time, FLOPs, or memory consumption between AR models/MLM models/diffusion models.
- The choice of tokenizers: The study dives deep into two tokenizers, VQGAN and BAE, and demonstrate the latter one’s effectiveness. However, the reason for choosing VQGAN is unclear and there are many other tokenizers that achieve much better performance than VQGAN with significantly better token utilization ratio.

**Methods And Evaluation Criteria:**

Appropriateness of Methods:
- The study explores multiple key design choices in tokenization, modeling, scanning patterns, vocabulary size, and sampling strategies, which are highly relevant for improving LLM-based image generation.
- Frechet Inception Distance (FID) is used as the main evaluation metric, which is a standard benchmark for generative models, making the comparison meaningful.
- Experiments are conducted on both 256×256 and 512×512 ImageNet, testing the scalability of the approach.

Potential Limitations:
- Lack of real-world benchmarks: Evaluations focus on class-conditional image generation on ImageNet, but no results are shown for text-to-image generation, which is a more practical setting. (This might not be a sufficient reason to reject this paper.)
- Computational efficiency is not discussed: While the study argues that AR models are effective, there is no analysis of training time, memory usage, or inference speed.

**Other Comments Or Suggestions:**

I will raise my score if the above questions are appropriately addressed.

**Other Strengths And Weaknesses:**

None

**Questions For Authors:**

None

**Relation To Broader Scientific Literature:**

- Language Models for Image Generation: The study builds on prior works applying LLMs to vision tasks (e.g., VQGAN, LlamaGen, VAR, MAR) but extends the exploration of architectural choices beyond existing models.
- Autoregressive vs. Masked Language Models: Consistent with NLP research, this paper finds that AR models scale better than MLMs, reinforcing prior insights from text-based LLMs.
- Comparison with Diffusion Models: The study follows research on diffusion-based image generation but suggests that AR models can achieve comparable quality under the right design choices.

**Theoretical Claims:**

Key Theoretical Contributions:
- The paper discusses scaling laws in AR models, showing that larger models capture both local and global information, leading to better image generation performance.
- It claims that MLMs struggle with image generation due to the lack of strict sequential dependencies in images, which aligns with prior findings in NLP.
- The study highlights the difference between text and image token distributions, suggesting that image tokens exhibit greater randomness than text tokens, which presents unique challenges for AR models.

Potential Issues:
- While the analysis of AR vs. MLM models is insightful, it lacks rigorous mathematical justification—there are no formal proofs to support the claim that AR is inherently better for image generation.
- The KL-divergence analysis of token distributions is interesting, but its impact on model optimization is not fully explored—how does randomness in tokenization affect convergence and learning stability?

---

> ### Author Rebuttal · Authors · 2025-04-01
>
> Thanks for your valuable comments! We will address your concerns in detail.
>
> ```Limit 1. no results on text-to-image generation```
>
> Our main goal is to assess the effectiveness of LLMs as a unified generation paradigm, especially when applied to images. To isolate modeling factors like tokenizers, AR vs. MLM, and sampling, we use **class-conditional generation** for its **simplicity and control**. While our findings (e.g., on scalable tokenization and AR modeling) are **relevant to text-to-image generation**, we consider T2I a **distinct challenge**—requiring further study on alignment, prompt structure, and retrieval—which requires delicate future work.
>
> ```Limit 2 & W1. Efficiency analysis```
>
> Our work focuses on understanding the **modeling behavior** and asses the **effectiveness** of LLMs for image generation. As LLMs acceleration techniques (e.g., KV-cache, FlashAttention, quantization) are well-developed and **directly transferable** to our setting, we do not emphasize efficiency in this paper. To address your concern, we summarize key efficiency insights below:
> * **Training efficiency**: AR is more efficient than MLM and diffusion. During training, each AR step predicts all tokens, while MLM predicts only a subset. As shown in Figure 2 in our main paper, AR converges in ~100 epochs, MLM in ~200, and DiT needs ~160 (refer to [Scalable Diffusion Models with Transformers]).
> * **Inference speed**: Vanilla AR is slower due to sequential decoding (e.g., 256 steps), but KV-cache accelerates it by ~4×. Summary: MLM > AR (with KV-cache) > Diffusion (DiT).
> * **Memory usage**: All models are **similar in training**. During inference, AR’s KV-cache increases memory usage with sequence length; MLM and diffusion require fewer steps and less memory per sample.
>
> We summarize FLOPs, training, and inference speed across AR, MLM, and DiT with matched model scales:
> |  | params(M) | FLOPs(G) | training epochs (converge/total) | inference time(sec/img) |
> |---|-----------|-----------|----------------------------------|------------------------|
> | DiT/XL-2 | 675 | 118.64 | ~160 / 1400 | 0.39 (50 steps) |
> | MLM-XL | 741 | 189.51 | ~200 / 400 | 0.10 (10 steps) |
> | AR-XL | 741 | 189.98 | ~100 / 400 | 0.15 (256 steps with KV-cache) |
>
> ``` Issue1. rigorous proof on AR&MLM in image domain```
>
> While we do not provide formal theoretical proof that AR is superior to MLM for image generation, our systematic empirical study offers **meaningful evidence** supporting this claim.
> To the best of our knowledge, **no existing work** in language or vision has established a rigorous theoretical justification. However, under our image-as-language framework, some NLP arguments for AR over MLM are partially **transferable**—e.g., AR benefits from **training-inference consistency**, while MLM suffers from **exposure bias** [1], and AR models the **full joint distribution**, unlike MLM [2].
>
> Moreover, the lack of standardized image tokenization adds complexity, making theoretical analysis more difficult and **beyond the scope** of this paper. We believe that our findings provide a **strong foundation** and **motivating evidence** for such future theoretical work.
>
> ``` Issue 2. The impact of KL-divergence analysis```
>
> We included the KL-divergence analysis as an **additional diagnostic tool** to investigate **why**, during next-token prediction on tokenized images, the **training loss remains high**—despite the model generating **high-quality images**, unlike in typical language modeling.  Actually, we conduct some analysis of the impact on model optimization in our paper.
> * Training loss curve **plateaus** rather than fully converging, as shown in **Figure 14 & 16 in Appendix**.  The KL-divergence analysis shows that image tokens are more random and less sequentially structured than language tokens, leading to less accurate predictions.
> * Attention Analysis: Attention maps analysis in Sec. 3.4 reveals strong local structure, and larger models capture more global dependencies, indicating **effective learning and scaling** despite the high loss.
>
> ```W2. Why choose VQGAN```
>
> We acknowledge the emergence of stronger tokenizers beyond VQGAN (e.g., hierarchical, residual, multi-scale) with improved utilization and reconstruction. However, we chose VQGAN as our baseline due to its **widespread use**, **reproducibility**, and **alignment with prior works** (e.g., MaskGIT, LDM, LlamaGen), enabling **controlled** comparisons. We include BAE as a **inherently** different tokenizer: unlike VQGAN’s **vector-level** quantization, BAE performs **bitwise scalar** quantization, allowing us to isolate the effect of granularity on the quantization. Its design is also compatible with other advanced quantization methods, suggesting potential for **future integration**.
>
> [1] Song et al., 2019. MASS: Masked Sequence to Sequence Pre-training for Language Generation.
>
> [2] Ghazvininejad et al., 2019. Mask-Predict: Parallel Decoding of Conditional Masked Language Models.

---

> > ### Comment · Reviewer_e5YQ · 2025-04-03
> >
> > The authors' rebuttal generally solves my question. Generally speaking, although without much rigorous proof, this work can provide useful priors for researchers to work on AR image generation, and it can contribute positively to the research community. I raise my score to 3.

---

> > > ### Author Response · Authors · 2025-04-07
> > >
> > > Thank you for your thoughtful consideration and for recognizing the value our work may bring to the community. We appreciate your assessment and support.

---

### Official Review · Reviewer_ED5q · 2025-03-14

**Overall Recommendation:** 3

**Summary:**

In this work, authors focus on the research topic of AR image generation, and conduct extensive studies focusing on 1) tokenizer, 2) AR model design (AR or Mask) 3) image scan direction. With large number of stduies, authors proposed ELM, which is able to achieve sota performance in ImageNet256 generation task.

**Claims And Evidence:**

Partly. Most of the claims are presented in related works.

**Essential References Not Discussed:**

See weakness about Infinity paper and LlamaGen paper

Infinity∞: Scaling Bitwise AutoRegressive Modeling for High-Resolution Image Synthesis

Autoregressive Model Beats Diffusion: Llama for Scalable Image Generation

**Experimental Designs Or Analyses:**

Yes. Please see the weakness and strengthes.

**Methods And Evaluation Criteria:**

Make sense.

**Other Comments Or Suggestions:**

Please see the weakness.

**Other Strengths And Weaknesses:**

Strengths:
Extensive experimental results demonstrate the effectiveness of proposed solution.


Weakness:
1. I have question about the results presented in Table 1. As shown in LlamaGen paper, 16k codebook VQGAN is able to achieve 2.19 rFID, trained with classical training receipe.  However, in the submission, only around 7.41. How to explain this significant performance gap?

2. Starts from line 171, authors claimed that "the introduction of Bernoulli Sampling during quantization improves generation performance". I think it is very closely related to Infinity paper  bitwise self-correction method. However, no discussions involoved.

3. I would suggest better re-design the Figure 2, the texts inside are too small.

4. For Scan Pattern Choice section, I was wondering if that makes sense since 1) raster is a common-practice choice 2) studies in Mamba related works have studied this choice and results indicate raster is simple and performs good 3) the common-practice raster choice also performs best as shown in Table 2.

5. I would say that  scaling law for AR image generation has already been validated by LlamaGen and VAR.

6. I have a question about the claim in Line 243 "When the vocabulary size exceeds a certain threshold, such as 2
16 (ie, 65,536), next-token prediction becomes significantly more challenging and may even become infeasible due to memory constraints." If that is true, how can we train LLMs? for example, the vocabulary size in Llama would be around 128k, and we can train Llama smoothly.

**Questions For Authors:**

Please check the weakness and looking forward to the rebuttal.

**Relation To Broader Scientific Literature:**

See weakness about Infinity paper and LlamaGen paper

Infinity∞: Scaling Bitwise AutoRegressive Modeling for High-Resolution Image Synthesis

Autoregressive Model Beats Diffusion: Llama for Scalable Image Generation

**Theoretical Claims:**

Not suitable.

---

> ### Author Rebuttal · Authors · 2025-04-01
>
> Thanks for your valuable comments! Below, we will answer your concern in detail.
>
> ```W1: Difference performance of VQGAN from LlamaGen```
>
> To ensure a **controlled comparison between structurally different quantization methods**, we use the **original VQGAN** from Taming Transformers [Esser et al., 2021], downloaded directly with rFID from the official repository （[taming-transformers-github](https://github.com/CompVis/taming-transformers)).
>
> LlamaGen tends to show that **discrete image tokenization** can achieve a reconstruction ability **close to continuous ones**. They **re-train VQGAN** with **L2-normalization on latent codes** refer to [1] to **improve codebook usage and representation smoothness**, which likely accounts for their stronger rFID results.
>
> To further address the reviewer’s concern, below table shows the **comparison between LlamaGen's VQGAN, the original Taming VQGAN, and BAE** (all with a down-sampling rate of 16 on 256px ImageNet):
> |     | VQ-16384 (taming) | VQ-16384 (LlamaGen) | BAE-2^16 | BAE-2^24 |
> |-------|-------------------|---------------------|-----------|-----------|
> |code usage| 8% |    97%               |  100%      |   100%      |
> | rFID     | 7.41             | 2.19                | 3.32      | 1.77      |
> | gFID(M.)  | 7.81         | 4.51                | 3.96      | 3.91      |
> | gFID(A.)   | 6.71         | 3.45                | 2.78      | 2.68      |
>
> This comparison shows that **BAE can still outperform both**,  ensuring **100% code utilization**, and the key advantage of **supporting larger codebooks** with better generation performance remains valid.
>
> ```W2. Discuss with Infinity paper - bitwise self-correction method. ```
>
> Both our method and Infinity∞ highlight that **introducing stochasticity or 'error' during training** —via Bernoulli sampling or bitwise self-correction—can improve **autoregressive image generation** by enhancing **robustness to prediction errors**. This suggests that **stochastic modeling** is a valuable direction for AR-based synthesis.
>
> The key difference lies in the **series modeling formulation**:
> * Infinity∞ builds on domain-specific serialization and **next-scale prediction**, where self-correction mitigates **error accumulation** across scales.
> * Our approach adopts **standard image tokenization** and treats images as language; Bernoulli sampling helps address the **inherent randomness** of image token sequences, improving **tolerance to uncertainty**.
>
> As Infinity∞ and our work were developed **concurrently**, we were not aware of it at the time of writing and therefore did not include a comparison. We appreciate the reviewer’s suggestion, and we will incorporate a discussion of this work in the revised version of our paper.
>
> ``` W3. We will re-design the Figures to improve the readability ```
>
> ```W4. Review on Scan Pattern Choice section```
>
> We study scan patterns to evaluate whether **generation order** impacts performance in the image-as-language setting, where token sequences lack inherent structure. Zigzag, commonly used in tasks like image compression and video coding, serves as a meaningful alternative to raster.
>
> Our results provide **empirical grounding** for choosing a raster scan in AR image generation, and the **minor performance differences** across scan patterns further suggest that large language models are **robust to token order variations**.
>
> ```W5. Discuss with the scaling law in LlamaGen and VAR```
>
> We agree that works like LlamaGen and VAR have demonstrated the scalability of AR models for image generation. Different from ours, VAR provides a **new** image autoregressive modeling mechanism. Our study is in line with LlamaGen, adopting a **language-centric** approach **without** domain-specific design, and offers complementary insights. Beyond generation quality scaling in LlamaGen, we further analyze **training dynamics, attention behavior, AR vs. MLM scaling trends**, and **the interactions between vocabulary and model capacity**, offering a broader perspective on what enables scalable image generation with LLMs. We believe our findings **complement and extend** those of LlamaGen and VAR and help build a more principled foundation for scaling image-as-language models.
>
> ```W6. question about the claim in Line 243, about the vocabulary threshold```
>
> We appologize that the statement in Line 243 is imprecise. Our intention was to highlight that as the vocabulary size approaches or exceeds **2^20**, the memory required for the output projection (softmax layer) in next-token prediction becomes a major bottleneck—especially in **image generation, where larger vocabularies are often needed for higher fidelity and resolution**.
>
> While LLMs like LLaMA handle vocab sizes up to ~200k (2^17) without issue, our use of 2^16 was meant as an **illustrative threshold**, not a hard limit. We will revise the wording for clarity in the final version.
>
>  [1] Yu et al. 2021, Vector-quantized image modeling with improved vqgan

---

> > ### Comment · Reviewer_ED5q · 2025-04-03
> >
> > Thank you for the rebuttal, which addressed some of my concerns.
> >
> > However, my primary concern remains: this work lacks methodological innovation and offers limited technical novelty, as also noted by Reviewer xmQD and Reviewer uJCp. The proposed methods are well-studied, and the conclusions drawn are largely well-established in existing literature.
> >
> > I am leaning toward a weak accept primarily due to the extensive experimental analysis. However, I am uncertain whether this alone is sufficient justification for acceptance. I will take into account further discussions with the other reviewers before finalizing my decision.
> >
> > Thank you.

---

> > > ### Author Response · Authors · 2025-04-07
> > >
> > > Thank you for your valuable feedback and for acknowledging the value of our experimental analysis. Our goal is to provide a systematic and comprehensive study that we believe offers meaningful insights for the community—especially as AR models are increasingly adopted as unified generative frameworks. We appreciate your consideration and hope the broader contribution of our work is useful in ongoing discussions.

---

### Decision · Program_Chairs · 2025-05-01

**Decision:**

Accept (poster)

**Comment:**

Even after the rebuttal and discussions, reviewers could not reach a unanimous verdict. The main remaining concern was the novelty of the paper, which most reviewers found weak. Despite this, the majority of the reviewers recommend accepting the paper, citing the thoroughness of the empirical findings. The rebuttal results also played a critical role in swaying the reviewers over the fence. The negative reviewer also acknowledges the impressiveness of these results and is ultimately okay with the paper being accepted. The AC, thus recommends accepting the paper.

The AC further strongly asks the authors to include the new results at the very least as an appendix to the paper, with the main manuscript pointing to the results, as they were quite important during the discussions.